# Pathogen effector recognition-dependent association of NRG1 with EDS1 and SAG101 in TNL receptor immunity

Xinhua Sun [1,8], Dmitry Lapin [1,2,8], Joanna M. Feehan[3,8], Sara C. Stolze [4], Katharina Kramer[4], Joram A. Dongus [1], Jakub Rzemieniewski [1,5], Servane Blanvillain-Baufumé[1], Anne Harzen[4], Jaqueline Bautor[1], Paul Derbyshire[3], Frank L. H. Menke [3], Iris Finkemeier [4,6], Hirofumi Nakagami [1,4], Jonathan D. G. Jones [3✉] & Jane E. Parker [1,7✉]

Plants utilise intracellular nucleotide-binding, leucine-rich repeat (NLR) immune receptors to detect pathogen effectors and activate local and systemic defence. NRG1 and ADR1 "helper" NLRs (RNLs) cooperate with enhanced disease susceptibility 1 (EDS1), senescence-associated gene 101 (SAG101) and phytoalexin-deficient 4 (PAD4) lipase-like proteins to mediate signalling from TIR domain NLR receptors (TNLs). The mechanism of RNL/EDS1 family protein cooperation is not understood. Here, we present genetic and molecular evidence for exclusive EDS1/SAG101/NRG1 and EDS1/PAD4/ADR1 co-functions in TNL immunity. Using immunoprecipitation and mass spectrometry, we show effector recognition-dependent interaction of NRG1 with EDS1 and SAG101, but not PAD4. An EDS1-SAG101 complex interacts with NRG1, and EDS1-PAD4 with ADR1, in an immune-activated state. NRG1 requires an intact nucleotide-binding P-loop motif, and EDS1 a functional EP domain and its partner SAG101, for induced association and immunity. Thus, two distinct modules (NRG1/EDS1/SAG101 and ADR1/EDS1/PAD4) mediate TNL receptor defence signalling.

---

[1] Department of Plant-Microbe Interactions, Max Planck Institute for Plant Breeding Research, Cologne, Germany. [2] Plant–Microbe Interactions, Department of Biology, Utrecht University, Utrecht, The Netherlands. [3] The Sainsbury Laboratory, University of East Anglia, Norwich, UK. [4] Proteomics group, Max Planck Institute for Plant Breeding Research, Cologne, Germany. [5] Department of Phytopathology, TUM School of Life Sciences Weihenstephan, Technical University of Munich, Freising, Germany. [6] Institute of Biology and Biotechnology of Plants, University of Muenster, Muenster, Germany. [7] Cologne-Düsseldorf Cluster of Excellence on Plant Sciences (CEPLAS), Düsseldorf, Germany. [8] These authors contributed equally: Xinhua Sun, Dmitry Lapin, Joanna M. Feehan. ✉email: jonathan.jones@TSL.ac.uk; parker@mpipz.mpg.de

Plants and animals have evolved structurally and functionally related cell surface and intracellular receptors that detect pathogen-derived molecules and activate innate immune responses. In both kingdoms, pathogen recognition by intracellular nucleotide-binding/leucine-rich repeat (NLR) receptors restricts disease[1]. Whereas mammals tend to have few functional NLR receptors, many plants have expanded and diversified NLR gene repertoires, likely in response to evolutionary pressure from host-adapted pathogens and pests[1,2]. Despite these different trajectories, plant and mammalian NLRs behave similarly as conformational switches for triggering defence and immune-related death pathways[3]. Plant NLRs directly bind pathogen strain-specific virulence factors (called effectors) or sense their modification of host immunity targets[4]. NLR-effector recognition leads to a process called effector-triggered immunity (ETI) which stops pathogen infection and is often accompanied by localised host cell death[5].

Increasing evidence in mammals and plants suggests NLR activation results from induced NLR oligomerization to form signalling-active scaffolds[6]. Plant NLR receptors are classified on the basis of their N-terminal signalling domain architectures: Toll/interleukin-1 receptor/resistance (TIR) NLRs (or TNLs) and coiled-coil (CC) NLRs (CNLs). The cryo-EM structure of a pathogen-activated CNL pentamer, *Arabidopsis* ZAR1, shows that five N-terminal domain protomers assemble a putative membrane-associated pore or channel which might represent a CC-mediated mechanism for activating defence signalling[7]. By contrast, structures of two pathogen-activated TNL receptor tetramers, *Arabidopsis* RPP1 and tobacco (*Nicotiana benthamiana*) Roq1, reveal that the four N-terminal TIR domains become reorganised to create a holoenzyme[8,9]. Studies show that TIR-domains have NAD$^+$ hydrolysis activity which, for plant TNLs, is necessary to initiate an authentic host immune response[10,11]. Hence, CNL- and TNL receptor early outputs appear to be different, though both are initiated by recognition-dependent oligomerization.

How NLR activation is transmitted to downstream pathways in ETI is more obscure, although CNLs and TNLs converge on qualitatively similar transcriptional programmes that drive local and systemic resistance[12–14]. NLRs also cooperate with cell surface pattern recognition receptor (PRR) systems mediating pattern-triggered immunity (PTI) to confer a fully effective immune response[15,16]. Moreover, CNLs and TNLs rely on a network of signalling NLRs (generically referred to as helper NLRs) to promote immunity and host cell death[17–19]. Two related sub-families of helper NLRs, N requirement gene 1 (NRG1)[20,21] and activated disease resistance 1 (ADR1)[22], are characterised by an N-terminal four-helix bundle domain with homology to *Arabidopsis* resistance to powdery mildew 8 (RPW8) and plant, fungal and mammalian mixed lineage kinase cell death executors (MLKLs) that have HET-S/LOP-B (HeLo) domains[23,24]. These helper NLRs are called CC$_R$-NLRs (or RNLs)[17]. In *Arabidopsis*, two functionally redundant *NRG1* paralogues (*NRG1.1* and *NRG1.2*) and three redundant *ADR1* paralogues (*ADR1*, *ADR1-L1* and *ADR1-L2*) contribute genetically to different extents to resistance and host cell death mediated by CNL and TNL receptors against a range of pathogens[12,19,25,26]. Functionally relevant interactions have not been found so far that would link RNLs molecularly to sensor NLRs or downstream signalling pathways.

The enhanced disease susceptibility 1 (EDS1) family of three lipase-like proteins, EDS1, senescence-associated gene 101 (SAG101) and phytoalexin deficient 4 (PAD4), constitutes a major NLR immunity signalling node[27]. *EDS1* is essential for TNL-dependent ETI across flowering plant species[26,28,29] and forms mutually exclusive, functional heterodimers with SAG101 or PAD4[30]. Genetic and biochemical characterisation of EDS1–SAG101 and EDS1–PAD4 dimers shows they have distinct functions in immunity[26,30–32]. EDS1–SAG101 appears to have coevolved with NRG1 group RNLs to signal specifically in TNL-triggered ETI[26]. By contrast, EDS1–PAD4, like ADR1 group RNLs, regulate a basal immunity response which, in *Arabidopsis*, slows virulent pathogen infection[12,22,25,28,32,33] and is utilised for ETI by TNL and CNL receptors[12,19,32]. A major role of *EDS1–PAD4* and *ADR1* RNLs in *Arabidopsis* basal immunity is to transcriptionally boost a genetically parallel salicylic acid (SA) phytohormone defence sector, which mediates local and systemic defences and is vulnerable to pathogen effector manipulation[28,33,34]. Recent studies revealed there is functional cooperation between EDS1–SAG101 and NRG1 RNLs in TNL ETI in *Arabidopsis* and *N. benthamiana*, consistent with their co-occurrence in angiosperm species[26,29,35]. Similarly, *Arabidopsis* *pad4* and *adr1*-family mutants phenocopy each other in various ETI and basal immunity responses[25,26]. Several groups have proposed that EDS1–SAG101 co-functions with NRG1s, and EDS1–PAD4 with ADR1s, thus constituting two distinct immunity signalling nodes downstream of NLR activation[21,25,26].

Here we present a genetic and biochemical characterisation of how *Arabidopsis* NRG1 and ADR1 RNLs co-function with EDS1 family members in NLR-triggered immunity. We show in *Arabidopsis* that *EDS1–SAG101–NRG1*s and *EDS1–PAD4–ADR1*s operate genetically as non-interchangeable signalling nodes in ETI. By performing immunoprecipitation and mass spectrometry analyses of *Arabidopsis* stable transgenic lines, we detect induced specific NRG1 association with EDS1 and SAG101 proteins and ADR1 association with EDS1 and PAD4 after TNL activation. We find in *Arabidopsis* stable transgenic lines and in *N. benthamiana* reconstitution assays that PTI activation is insufficient for NRG1-induced association with EDS1 and SAG101. Both the EDS1–SAG101 heterodimer and NRG1 are necessary to form a signalling-active protein complex. Our data provide a molecular insight to two functionally different RNL signalling nodes operating with specific EDS1-family members to mediate defences downstream of NLR receptor activation.

## Results

**Distinct *PAD4-ADR1* and *SAG101–NRG1* modules in TNL$^{RRS1-RPS4}$ immunity.** We tested in *Arabidopsis* whether individual components of the proposed *EDS1–PAD4–ADR1*s and *EDS1–SAG101–NRG1*s immunity modules[25,26] are genetically interchangeable. We reasoned that the replacement of *ADR1* by *NRG1* group members, and reciprocally *PAD4* by *SAG101*, would reveal cross-utilization of components. Combinations of previously characterised *Arabidopsis* EDS1 family mutants (*pad4*, *sag101*, and *pad4 sag101*) with *ADR1* (*adr1 adr1-L1 adr1-L2*, denoted *a3*)[22] or *NRG1* group (*nrg1.1* and *nrg1.2*; denoted *n2*)[26] mutants were generated in accession Col-0 (Col). This produced mutant groups I, II and III (Fig. 1a), with group III containing between-module combinations. Group I, II and III mutants were tested for resistance mediated by the TNL pair RRS1 and RPS4 (TNL$^{RRS1–RPS4}$) to *Pst avrRps4* infection in leaves, measured against wild-type Col-0 (Col, resistant), an *rrs1a rrs1b* (*rrs1ab*) mutant defective specifically in TNL$^{RRS1–RPS4}$ ETI[36], as well as *eds1* and an *a3 n2* 'helperless' mutant which are both fully susceptible to *Pst avrRps4*[12,19,25,26]. In *Pst avrRps4* bacterial growth assays, *pad4 a3* phenocopied the partial resistance of *pad4* and *a3* single mutants, and *sag101 n2* phenocopied *sag101* and *n2* full resistance (Fig. 1b, c). The cross-pathway *pad4 n2* and *sag101 a3* combinations in Group III were as susceptible as *eds1*, *pad4 sag101* and *a3 n2* mutants (Fig. 1b, c). We further tested group I, II and III mutants for host cell death responses to a type

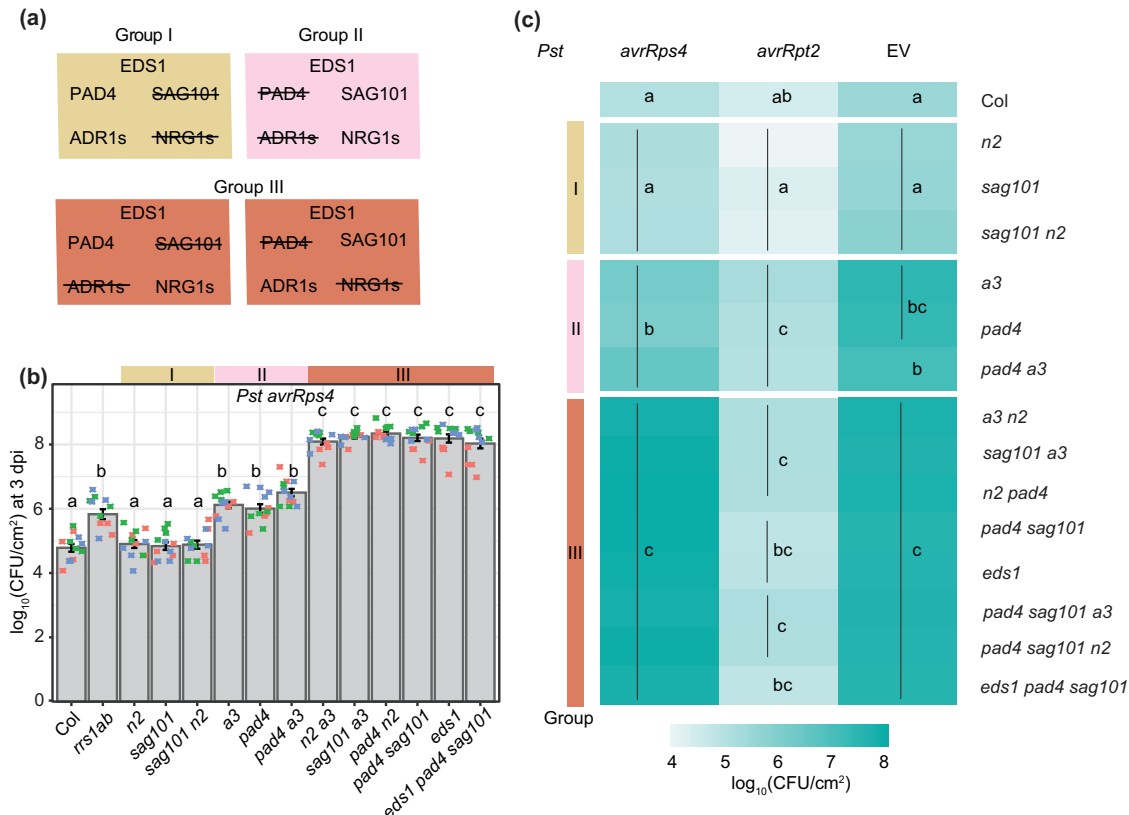

**Fig. 1 Distinct *PAD4–ADR1* and *SAG101–NRG1* modules operate in *Arabidopsis* TNL^RRS1–RPS4 immunity. a** Overview of mutants used in (**b**) and (**c**). Group I comprises mutants disabled in *SAG101* and/or *NRG1*s: *sag101*, *nrg1.1 nrg1.2* (*n2*) and *sag101 n2*. Group II has mutants in *PAD4* and/or *ADR1*s: *pad4*, *adr1 adr1-L1 adr1-L2* (*a3*) and *pad4 a3*. Group III is composed of cross-branch combinatorial mutants *a3 n2*, *sag101 a3*, *pad4 n2*, *pad4 sag101*, *sag101 pad4 a3*, *sag101 pad4 n2*, *eds1 pad4 sag101*. **b** Growth of *Pseudomonas syringae* pv. *tomato* DC3000 (*Pst*) *avrRps4* in leaves of *Arabidopsis* Col-0 (Col) and indicated mutants at 3 days post inoculation (dpi) via syringe infiltration (OD_600 = 0.0005). Bacterial loads are shown as $\log_{10}$ colony-forming units (CFU) per $cm^2$. Experiments were performed three times independently with four replicates each (Tukey's HSD, $\alpha = 0.001$, $n = 12$). Error bars represent standard error of mean. Datapoints with the same colour come from one independent experiment. **c** Growth of *Pst avrRps4*, *Pst avrRpt2* or *Pst* (empty vector, EV) in indicated *Arabidopsis* lines at 3 dpi infected via syringe infiltration (OD_600 = 0.0005). Heatmap represents mean $\log_{10}$-transformed CFU values from three (*Pst avrRps4*), four (*Pst*) or five (*Pst avrRpt2*) independent experiments, each with four replicates ($n = 12$ for *Pst avrRps4*, $n = 16$ for *Pst*, $n = 20$ for *Pst avrRpt2*). Statistical significance codes are assigned based on Tukey's HSD ($\alpha = 0.001$). *sag101 a3* and *pad4 n2* phenocopy *pad4 sag101* and *a3 n2*, indicating that SAG101 does not form functional signalling modules with ADR1s, and NRG1s with PAD4. Note: significance codes are assigned based on the statistical analysis per treatment and should be read columnwise. The jitter plot in **b** shows individual data points used to calculate means on the heatmap for *Pst avrRps4* infection. *Pst Pseudomonas syringae* pv. *tomato* DC3000, CFU colony-forming units, EV empty vector.

III secretion system-equipped *P. fluorescens* strain *Pf*0-1 delivering *avrRps4*, visually at 24 h post infiltration (hpi) (Supplementary Fig. 1a) and by quantitative electrolyte leakage assays over 6–24 hpi (Supplementary Fig. 1b–e). This produced the same phenotypic clustering of mutants as the *Pst avrRps4* resistance assays—*pad4* and *a3* behaved like *pad4 a3*, while defects in *sag101* and *n2* aligned with those in *sag101 n2*. Put together, these data show that there is exclusive cooperation between *PAD4* and *ADR1* RNLs in a single pathway leading to restriction of bacterial growth, and between *SAG101* and *NRG1* RNLs in promoting host cell death and resistance in TNL^RRS1–RPS4 immunity. Furthermore, the data argue strongly against physiologically relevant cross-utilization of components between the *PAD4–ADR1*s and *SAG101–NRG1*s signalling modules.

**The *EDS1–SAG101–NRG1* node is dispensable for CNL^RPS2–dependent ETI.** Resistance to *Pst avrRpt2* mediated by the CNL receptor RPS2 was compromised only when *PAD4* or *ADR1* RNLs were mutated in group I, II and III mutants, and there was no measurable contribution of *SAG101* or *NRG1* RNLs to RPS2

immunity, even in a *pad4 a3* background (Fig. 1c, Supplementary Fig. 2a). In quantitative electrolyte leakage assays, we detected equivalent contributions of *PAD4* and *ADR1* RNLs to RPS2 triggered host cell death at 6 and 8 hpi, but not later at 24 hpi (Supplementary Fig. 3), as seen previously for *pad4* and *eds1* mutants[33] and *a3*[12]. Importantly, *PAD4* and *ADR1*s early promotion of RPS2 cell death could not be substituted by *SAG101* or *NRG1*s in any of the mutant lines (Supplementary Fig. 3a, b). We concluded that EDS1-PAD4 also work together with ADR1s in a single pathway to promote CNL^RPS2-triggered early host cell death and that *SAG101* and *NRG1*s are not recruited for CNL^RPS2 ETI, even when *PAD4* and *ADR1*s are disabled.

Next, we investigated whether recruitment of *PAD4* and *ADR1*s and apparent non-utilization of *SAG101* and *NRG1*s in RPS2-triggered immunity is masked by the genetically parallel *ICS1*-dependent SA defence hormone pathway[22,32,33,37]. For this, we introduced an *ics1* (*sid2*) mutation into the single-module (*pad4 a3*, *sag101 n2*) and cross-module (*pad4 n2* and *sag101 a3*) mutant backgrounds (Fig. 2a). Loss of *ICS1* did not alter the different dependencies of TNL^RRS1–RPS4 or CNL^RPS2 on *PAD4–ADR1*s and *SAG101–NRG1*s in bacterial resistance

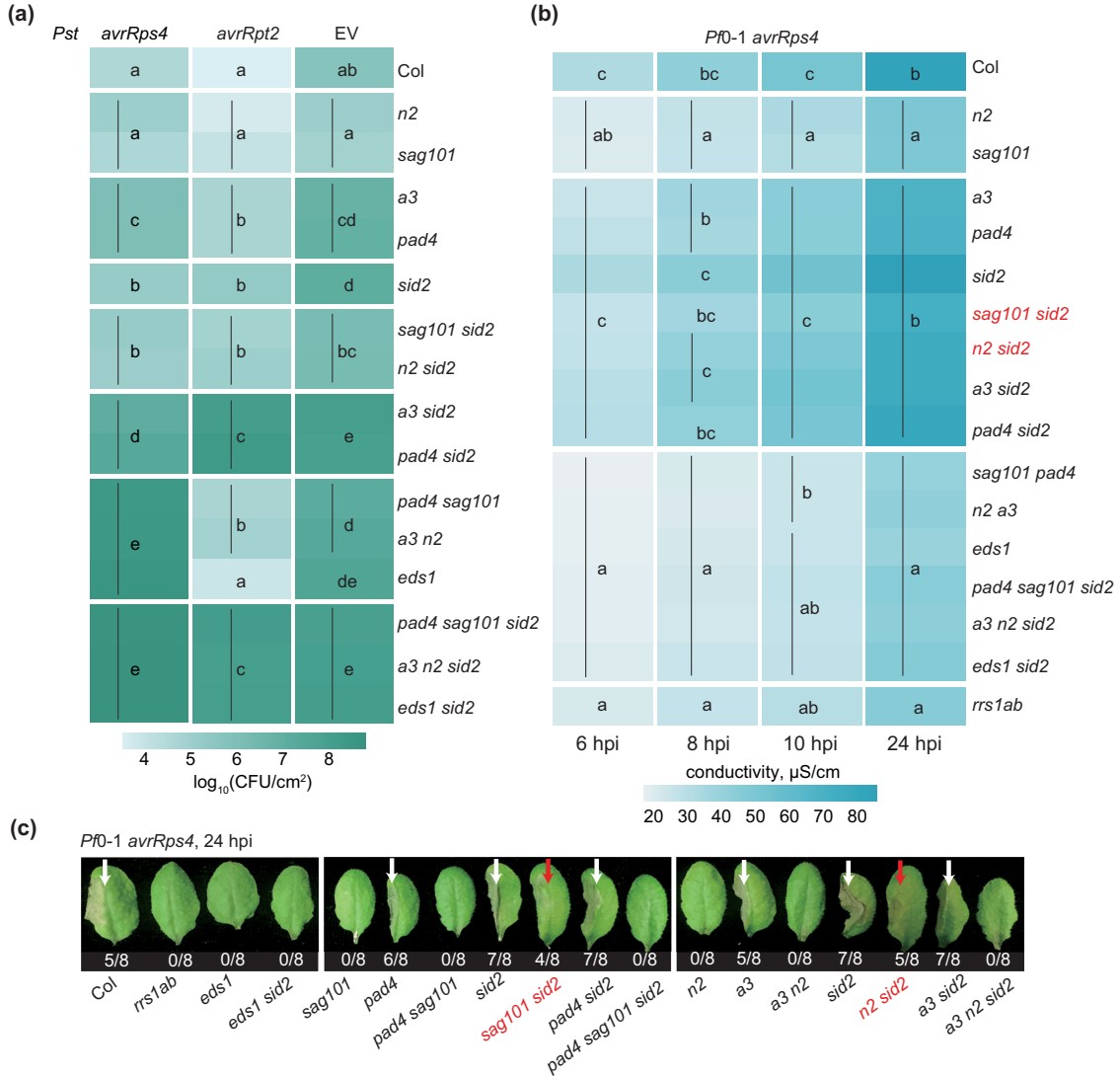

**Fig. 2 Removal of the *SAG101–NRG1* and *ICS1* sectors reveals *PAD4–ADR1*-promoted TNL cell death. a** A heatmap representation of *Pst avrRps4*, *Pst avrRpt2* or *Pst* (empty vector, EV) growth at 3 dpi in leaves of indicated genotypes (syringe infiltration, $OD_{600} = 0.0005$). The significance codes are based on Tukey's HSD test ($\alpha = 0.001$, $n = 12$ for *Pst avrRps4* and *Pst avrRpt2*, $n = 16$ for *Pst*). Data points were combined from three (*Pst avrRps4*, *Pst avrRpt2*) or four (*Pst*) independent experiments, each with four replicates. Note: significance codes are assigned based on the statistical analysis per treatment and should be read columnwise. **b** A heatmap of quantitative cell death assays conducted on leaves of indicated genotypes after infiltration utilising the *Pseudomonas fluorescens* 0–1 effector tester strain (hereafter *Pf*0-1) delivering avrRps4 ($OD_{600} = 0.2$). Cell death was measured by electrolyte leakage from bacteria-infiltrated leaf discs at 6, 8, 10 and 24 hpi. Data are displayed as means from four experiments, each with four replicates ($n = 16$). Statistical significance codes for the difference in means are based on Tukey's HSD test ($\alpha = 0.001$). In mutants marked in red, the *PAD4–ADR1*s cell death branch operates in TNL$^{RRS1-RPS4}$ immunity when *SAG101–NRG1*s and *ICS1* pathways are not functional. Note: significance codes are assigned based on the statistical analysis per timepoint and should be read columnwise. Boxplot representation of the same data is provided in Supplementary Figs. 5 and 6. **c** Visual cell death symptoms at 24 hpi *Pf*0-1 avrRps4 infiltrating into leaf halves of indicated genotypes as in **b**. The ratio beneath each leaf indicates number of leaves with visible tissue collapse from all infiltrated leaves in two independent experiments. White arrows mark cell death visible as tissue collapse in a manner dependent on *SAG101–NRG1*s. Red arrows mark cell death in lines without functional *SAG101–NRG1*s and *ICS1*. Genotypes in red have functional *PAD4–ADR1*s but not *SAG101–NRG1*s. *Pst Pseudomonas syringae* pv. *tomato* DC3000, *Pf*0-1 effector tester strain of *Pseudomonas fluorescens* 0–1, CFU colony-forming units, EV empty vector.

(Fig. 2a; Supplementary Fig. 4a–d). We concluded that *PAD4–ADR1*s and *SAG101–NRG1*s distinctive contributions to resistance mediated by these TNL and CNL receptors are independent of *ICS1*-generated SA.

We further tested whether a possible *SAG101* and *NRG1*s role in basal resistance against virulent *Pst* (Fig. 1c, Supplementary Fig. 2b)[12,25] is redundant with and therefore obscured by the SA and *PAD4-ADR1*s sectors[13,22,38]. We found that *sag101* and *n2* mutations did not increase *sid2*, *sid2 pad4* and *sid2 a3* susceptibility to *Pst* (Fig. 2a, Supplementary Fig. 4e, f). Therefore,

*SAG101* and *NRG1*s do not contribute to *Arabidopsis* basal resistance to *Pst* bacteria in a susceptible interaction.

**TNL cell death via *PAD4–ADR1*s in the absence of *SAG101–NRG1*s and *ICS1*.** *SAG101* and *NRG1*s are dispensable for TNL$^{RRS1-RPS4}$ resistance unless *PAD4* and/or *ADR1*s are disabled (Figs. 1b, 2a; Supplementary Fig. 4a, b), consistent with unequal contributions of these two branches in TNL immunity[12,19,25,26]. We used the *Arabidopsis* combinatorial mutants with *sid2* (Fig. 2a) to explore whether *ICS1*-synthesised

SA affects *PAD4–ADR1*s or *SAG101–NRG1*s involvement in TNL$^{RRS1-RPS4}$ triggered host cell death. In quantitative electrolyte leakage assays over 6–24 hpi and macroscopically at 24 hpi, *pad4*, *a3*, *sid2*, *pad4 sid2* and *a3 sid2* mutants displayed similar leaf cell death responses as wild-type Col (Fig. 2b, c; Supplementary Figs. 5 and 6). Therefore, *PAD4* and *ADR1*s are dispensable for TNL$^{RRS1-RPS4}$ cell death, regardless of *ICS1*-dependent SA status. While *n2* and *sag101* mutants had strongly reduced host cell death, as expected[12,19,26], we observed an early cell death response similar to that of wild-type Col in *sag101 sid2* and *n2 sid2* backgrounds (Fig. 2b, c; Supplementary Figs. 5 and 6). This restored cell death was abolished in *pad4 sag101 sid2* and *a3 n2 sid2* mutants (Fig. 2b, c; Supplementary Figs. 5 and 6). We concluded that an *EDS1–PAD4–ADR1*s controlled mechanism can lead to host cell death in TNL$^{RRS1-RPS4}$ immunity that is likely antagonised or restricted by combined *EDS1–SAG101–NRG1*s and SA functions.

Previously, SA was found to conditionally suppress leaf cell death promoted by metacaspase 1 (MC1) in CNL RPM1 ETI[39], and *MC1*, *PAD4* or *ADR1*s promoted runaway cell death caused by the loss of *Lesion Simulating Disease1* (*LSD1*)[22,40–42]. Therefore, we tested whether MC1 is required for the *PAD4–ADR1*s-dependent TNL$^{RRS1-RPS4}$ cell death that was restored in *sag101 sid2* (Fig. 2b, c; Supplementary Figs. 5 and 6). For this, we generated a *sag101 sid2 mc1* triple mutant and measured its cell death phenotype alongside *mc1* and *pad4 sid2 mc1* lines. The *mc1* mutation did not compromise *SAG101* or *PAD4* promoted TNL cell death (Supplementary Fig. 7), suggesting that *MC1* is dispensable for both *SAG101–NRG1*s- and *PAD4–ADR1*s-driven cell death responses in TNL-triggered bacterial immunity. Collectively, these data show that *EDS1–SAG101–NRG1*s and *EDS1–PAD4–ADR1*s are genetically hard-wired signalling modules in NLR immunity that react differently to *ICS1*-generated SA.

**PAD4 and SAG101 interact, respectively, with ADR1s and NRG1s.** We tested whether the genetic co-requirement of *EDS1–PAD4–ADR1*s and *EDS1–SAG101–NRG1*s results from specific molecular associations. We generated complementing *pPAD4:YFP-PAD4* and *pSAG101:SAG101-YFP* stable transgenic lines in a Col *pad4 sag101* double mutant background (Supplementary Fig. 8) to test whether each EDS1 partner associates with similar or different RNL proteins in the TNL ETI response. The double *pad4 sag101* mutant background of the *pSAG101:SAG101-YFP* complementation line allows *SAG101–NRG1*s to engage in cell death and pathogen resistance conferred by RRS1–RPS4)[26] (Supplementary Fig. 8) and also in transcriptional reprogramming that is otherwise chiefly controlled by *PAD4-ADR1*s in TNL$^{RRS1-RPS4}$ ETI[12,32]. These two lines and a Col *p35S:YFP-StrepII-3xHA* (YFP-SH) control were infiltrated with *Pf*0-1 *avrRps4*. *Pf*0-1 *avrRps4*-elicited leaf total soluble extracts were processed at 6 hpi because *EDS1*-dependent transcriptional reprogramming starting at ~4 hpi is critical for RRS1–RPS4 resistance[12,13,32,43]. SAG101-YFP, YFP-PAD4 and YFP-SH proteins were purified via immunoprecipitation (IP) with GFP-trap agarose beads. Liquid chromatography and mass spectrometry (MS) (LC–MS) analyses showed strong enrichment of the two Col-0 native EDS1A and EDS1B isoforms in both SAG101-YFP or YFP-PAD4 samples (Fig. 3a; Supplementary Data 2), as expected from earlier studies[30,31,44,45]. EDS1A and EDS1B were also detected at a low level in YFP-SH control IPs (Fig. 3a), consistent with EDS1 weak non-specific interaction when its direct partners (PAD4 and SAG101) are missing[45,46]. NRG1.1 and NRG1.2 peptides were highly enriched in SAG101-YFP but not YFP-PAD4 or YFP-SH samples. Notably, peptides derived

from NRG1.3, a truncated NRG1 isoform that does not contribute genetically to TNL ETI responses[25], were enriched in SAG101-YFP IPs, and less strongly with YFP-PAD4 (Fig. 3a; Supplementary Fig. 9; Supplementary Data 2). By contrast, ADR1-L1 and ADR1-L2 co-purified with YFP-PAD4 but were not detected in SAG101-YFP or YFP-SH IP samples (Fig. 3a; Supplementary Fig. 9; Supplementary Data 2), suggesting that ADR1 group RNLs interact preferentially with PAD4 over SAG101 at 6 hpi. A number of other immunity components were differentially enriched by SAG101-YFP and PAD4-YFP in the LC–MS analysis (Supplementary Fig. 9b), consistent with the two EDS1 partners contributing to different processes in TNL$^{RRS1-}$$^{RPS4}$ immunity. Put together, the *Arabidopsis* IP-MS analyses show that EDS1–PAD4 and EDS1–SAG101 dimers interact specifically with RNLs in TNL-induced tissues at 6 hpi. The observed preferential association of PAD4 with ADR1-L1 and ADR1-L2, and SAG101 with NRG1.1 and NRG1.2, further suggests that EDS1–PAD4 and EDS1–SAG101 complexes with specific helper RNL types underpin these genetically distinct *Arabidopsis* immunity modules promoting host resistance and cell death responses in TNL ETI[19,32] (Figs. 1 and 2).

**Early effector-dependent NRG1 association with EDS1 and SAG101.** To investigate whether the RNLs associate with EDS1, we enriched for EDS1 protein from a transgenic Col *eds1* line expressing *pEDS1:EDS1-YFP*[43] at 8 h after infiltrating leaves with *Pst avrRps4* bacteria and preparing nuclear-enriched extracts. Garcia et al. (2010) demonstrated the importance of an EDS1 nuclear pool for gene expression reprogramming in TNL$^{RRS1-RPS4}$ triggered ETI[43]. To interrogate protein associations with EDS1, we immunoprecipitated EDS1-YFP using GFP-trap agarose beads and analysed co-purified proteins via LC–MS. GFP-trap purification and LC–MS processing of eluates from a Col mutant in *Telomere Repeat Binding 1* (*TRB1*) expressing nuclear localised TRB1-GFP[47] was used as a control for non-specific associations (Fig. 3b; Supplementary Fig. 10; Supplementary Data 3). PAD4 and SAG101 were highly enriched in EDS1-YFP relative to TRB1-GFP pulldowns, as represented in a volcano plot (Supplementary Fig. 10), and consistent with EDS1 stable PAD4 or SAG101 dimers persisting in the TNL$^{RRS1-RPS4}$ ETI response[44,45]. NRG1.1 and NRG1.2 proteins were also specifically enriched in EDS1-YFP compared to TRB1-GFP samples (Fig. 3b). We did not detect any of the three functional *Arabidopsis* ADR1 isoforms, ADR1, ADR1-L1 and ADR1-L2[22] (Fig. 3b; Supplementary Data 3). In the nuclei-enriched protein extracts separated by native size exclusion chromatography, EDS1-YFP eluted between ~50 and ~600 kDa (Fig. 3c). An EDS1-YFP peak at ~160 kDa with a higher molecular weight tail is consistent with EDS1 forming stable exclusive heterodimers with PAD4 or SAG101[30,31,45] and sub-stoichiometric higher-order complexes (Fig. 3c). Together, these data suggest that NRG1.1 and NRG1.2 also interact with EDS1 in TNL$^{RRS1-RPS4}$-activated cells.

We tested whether the association of NRG1 group RNLs with EDS1 and SAG101 is dependent on TNL$^{RRS1-RPS4}$ activation using an *Arabidopsis* Ws-2 *n2* complementation line expressing *pNRG1.2:NRG1.2-6xHis-3xFLAG* (NRG1.2-HF)[19]. Leaves were infiltrated with *Pf*0-1 bacteria delivering avrRps4 (*Pf*0-1 *avrRps4*) to activate RRS1–RPS4 ETI or *Pf*0-1 with an empty vector (EV) as a negative control eliciting PTI[48,49]. Soluble extracts from *Pf*0-1 EV and *Pf*0-1 *avrRps4*-infiltrated leaves at 4 hpi were processed to monitor early changes during the *EDS1*-dependent transcriptional reprogramming in TNL$^{RRS1-RPS4}$ resistance[12,13,32,43,49]. After NRG1.2-HF immunopurification on α-FLAG agarose beads, LC–MS analysis revealed spectra for peptides derived from EDS1 and SAG101 in

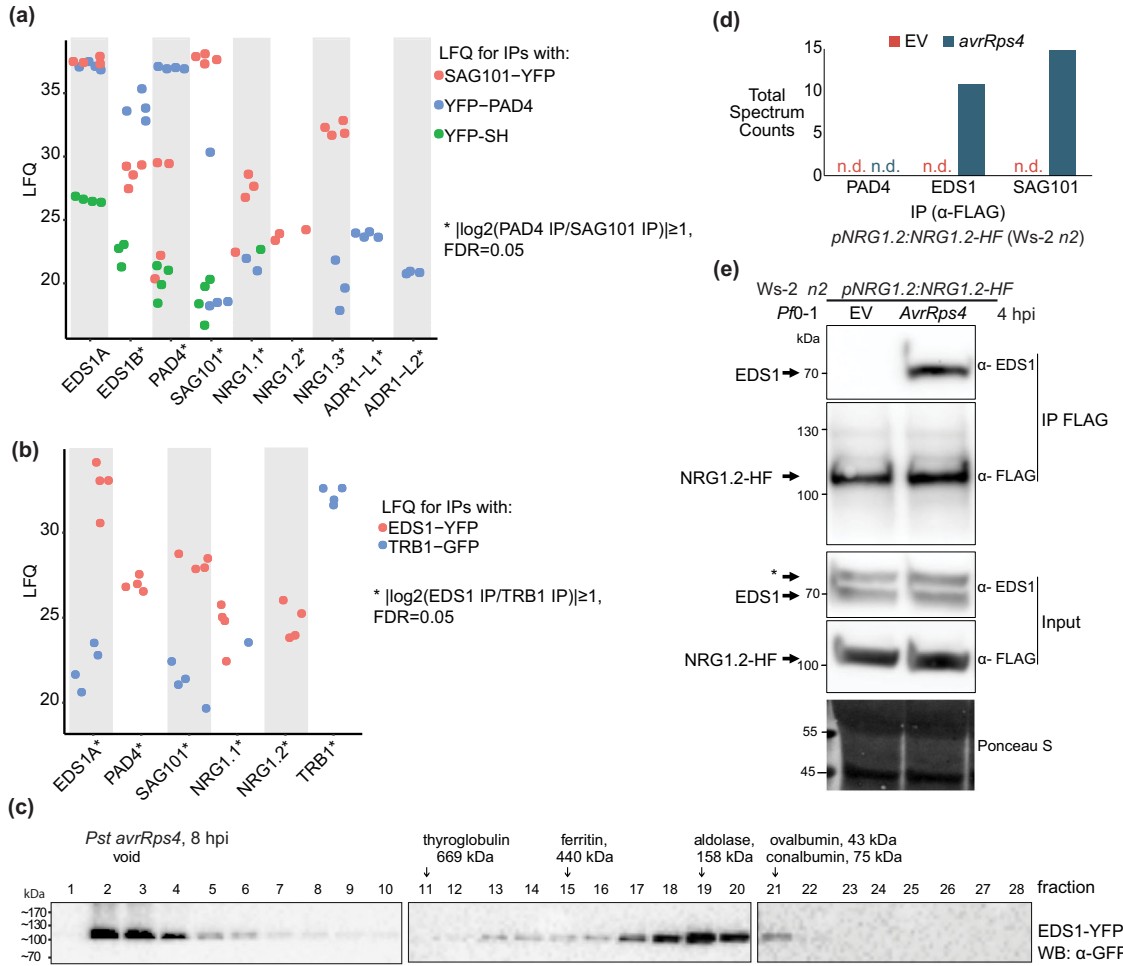

**Fig. 3 Early effector-dependent NRG1 association with EDS1 and SAG101 in *Arabidopsis* TNL^RRS1–RPS4 triggered ETI. a** Dotplot of normalised abundance values (label-free quantification (LFQ), log₂-scaled) for proteins detected in liquid chromatography–mass spectrometry (LC–MS) analyses after α-GFP immunoprecipitation (IP) of PAD4-YFP or SAG101-YFP from total leaf extracts of respective complementation lines *pPAD4:YFP-PAD4* and *pSAG101: SAG101-YFP* (both Col *pad4 sag101* background) after infiltration with effector tester strain of *Pseudomonas fluorescens* 0-1 (*Pf*0-1) *avrRps4* (6 hpi, OD₆₀₀ = 0.2). ADR1s are specifically enriched in PAD4-YFP IP samples, whereas NRG1s are more abundant in the SAG101-YFP IP samples. Samples were collected from four independent experiments. **b** Dotplot of LFQ values for proteins detected in LC–MS analyses after IP of EDS1-YFP and TRB1-GFP from nuclei-enriched extracts of corresponding *Arabidopsis* complementation lines infiltrated with *Pseudomonas syringae* pv. *tomato* DC3000 (*Pst*) *avrRps4* (8 hpi, OD₆₀₀ = 0.1). NRG1.1 and NRG1.2 are specifically enriched in EDS1-YFP samples. Samples were collected from four independent experiments. **c** α-GFP probed immunoblots of nuclei-enriched extracts from leaves of the *Arabidopsis pEDS1:EDS1-YFP* complementation line (Col *eds1-2* background) infiltrated with *Pst avrRps4* (OD₆₀₀ = 0.1, 8 hpi). Extracts were resolved using native gel filtration. Arrows below protein markers indicate position of the corresponding peak. Numbers refer to column fractions. EDS1 forms stable ~100–600 kDa complexes. The experiment was conducted three times with similar results. **d** LC–MS analysis of eluates after α-FLAG IP of total leaf extracts from *Arabidopsis* Ws-2 *n2 pNRG1.2:NRG1.2-HF* complementation line infiltrated with *Pf*0-1 empty vector (EV) or *Pf*0-1 *avrRps4* (4 hpi, OD₆₀₀ = 0.3). Peptides corresponding to EDS1 and SAG101 were observed in eluates only after *Pf*0-1-mediated delivery of avrRps4. This result was observed in two independent experiments. **e** Western blot (WB) analysis of eluates from (**d**). Asterisk indicates a nonspecific band on the α-EDS1 blot for input samples. The analysis was performed on total leaf extracts and was conducted four times with similar results. Association of EDS1 with NRG1.2-HF was observed only after *Pf*0-1-mediated delivery of avrRps4. Ponceau S staining shows similar protein loading in input samples on the blot.

elution products isolated from *Pf*0-1 *avrRps4* but not *Pf*0-1 (EV)-treated tissues (Fig. 3d). No PAD4 peptides were identified in the analysis (Supplementary Data 4). Elution products resolved by SDS–PAGE and probed with α-FLAG and α-EDS1 antibodies also revealed an association of EDS1 with NRG1.2 that is dependent on *Pf*0-1 delivery of avrRps4 (Fig. 3e). Importantly, no association between NRG1.2 and EDS1 or SAG101 was detected at 4 hpi with *Pf*0-1 alone, indicating that a PTI response[15,16,50] is insufficient to induce an NRG1 interaction with EDS1 and SAG101 (Fig. 3d, e; Supplementary Data 4). These data show that TNL^RRS1–RPS4 activation induces NRG1.2 associations with EDS1 and SAG101, but not PAD4, in *Arabidopsis*.

**NRG1.1 interacts with EDS1 and SAG101 after tobacco TNL activation.** Previously, transient co-expression of *Arabidopsis* EDS1, SAG101 and NRG1.1 or NRG1.2 proteins in an *N. benthamiana* CRISPR quadruple *eds1a pad4 sag101a sag101b* (*Nb epss*) mutant reconstituted host cell death and bacterial resistance after TNL^Roq1 recognition of the bacterial effector XopQ[26,51]. We exploited the *Nb epss* transient assay system to investigate molecular requirements for *Arabidopsis* EDS1, SAG101, NRG1s associations and functions in ETI. For this, we developed an ETI assay in *Nb epss* leaves with more precise timing of the TNL^Roq1-dependent response than previously achieved using only *Agrobacteria*-mediated expression

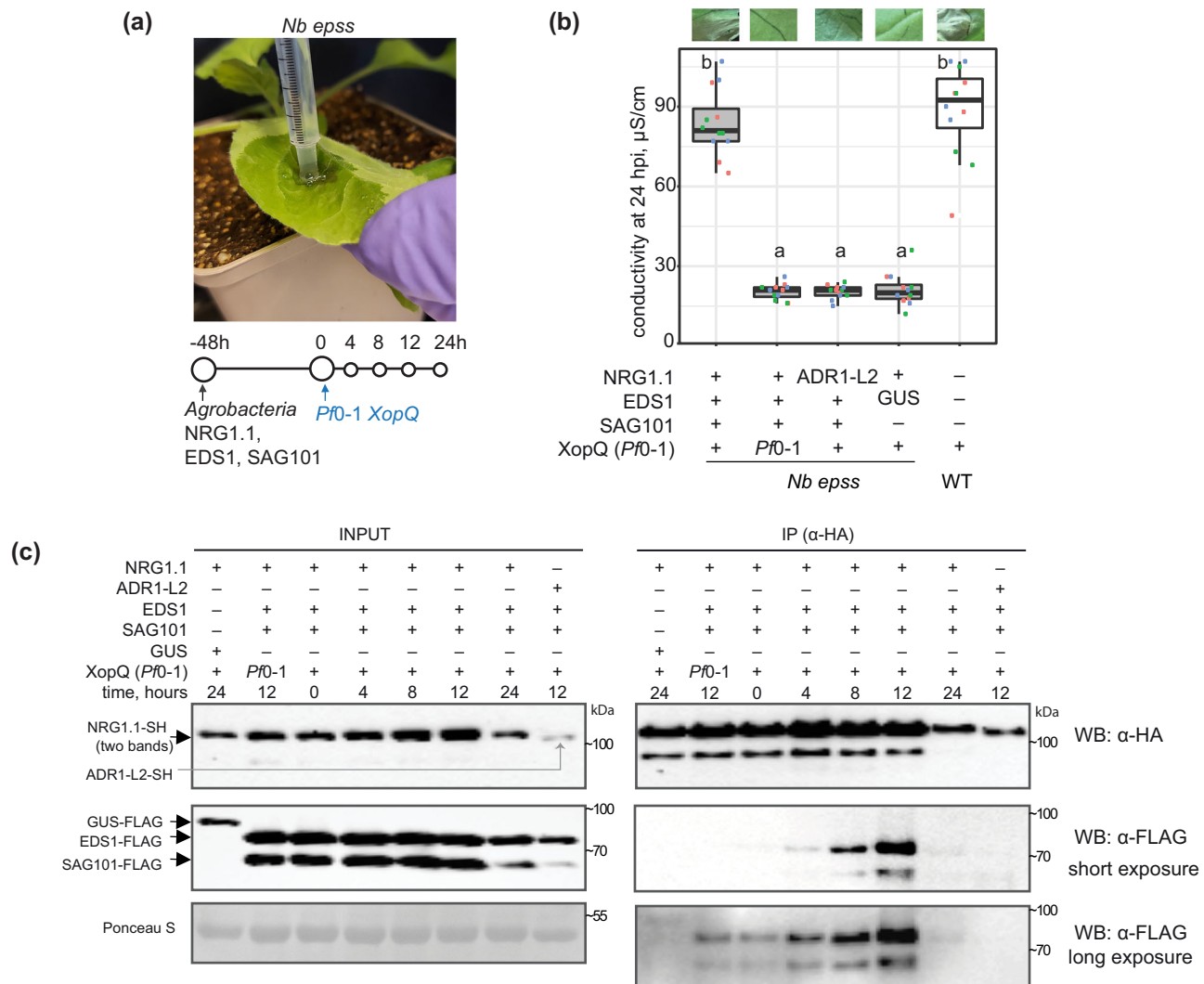

**Fig. 4 XopQ/Roq1-dependent *Arabidopsis* NRG1.1 association with EDS1 and SAG101 in *Nicotiana benthamiana*. a** Sample collection scheme for experiments in (**b**) and (**c**). Roq1-dependent cell death is restored in *Nicotiana benthamiana* (*Nb*) *eds1 pad4 sag101a sag101b* (*Nb epss*) signalling deficient mutant by expressing *Arabidopsis* EDS1, SAG101 and NRG1.1 via *Agrobacteria* infiltration 48 h before XopQ effector delivery. *Pf*0-1 *XopQ* is syringe-infiltrated ($OD_{600} = 0.3$) to deliver the effector and to study TNL signalling events in a time-resolved manner up to 24 hpi. **b** Macroscopic symptoms and quantification of XopQ-triggered cell death at 24 hpi after infiltrating *Pf*0-1 *XopQ* in leaves of *Nb epss* expressing *Arabidopsis* EDS1-FLAG, SAG101-FLAG, NRG1.1-3xHA-StrepII (NRG1.1-SH) or ADR1-L2-3xHA-StrepII (ADR1-L2-SH). *Pf*0-1 with empty vector (*Pf*0-1) served as a "no-ETI" control. Cell death was quantified in electrolyte leakage assays 6 h after harvesting leaf discs (24 hpi with *Pf*0-1 *XopQ*). The experiment was performed three times independently, each with four replicates (leaf discs) (Tukey's HSD, $\alpha = 0.001$, $n = 12$). Transiently expressed *Arabidopsis* EDS1, SAG101 and NRG1.1 proteins are functional in *Pf*0-1 *XopQ* triggered (Roq1) cell death. Description of boxplots: minima—first quartile, maxima—third quartile, centre—median, whiskers extend to the minimum and maximum values but not further than 1.5 interquartile range from the respective minima or maxima of the boxplot. Datapoints with the same colour come from one independent experiment. **c** Coimmunoprecipitation assay followed by Western blotting to test for XopQ-triggered associations between *Arabidopsis* NRG1.1-SH and FLAG-tagged EDS1 or SAG101 in *Nb epss* according to the infiltration scheme in (**a**). NRG1.1-SH or ADR1-SH were enriched using α-HA agarose beads, and presence of FLAG-tagged EDS1, SAG101 or GUS was tested by probing blots with α-FLAG antibodies. IP assays were conducted three times independently with similar results. NRG1.1 association with EDS1 and SAG101 requires Roq1/XopQ ETI activation. *Pf*0-1 effector tester strain of *Pseudomonas fluorescens* 0–1 with empty vector, *Pf*0-1 *XopQ* (in **a**) or XopQ (*Pf*0-1) effector tester strain of *Pseudomonas fluorescens* 0–1 delivering *XopQ*, WB Western blotting, IP immunoprecipitation, *Nb epss Nicotiana benthamiana eds1a pad4 sag101a sag101b*.

of various components[26]. In the modified assay, we transiently expressed combinations of epitope-tagged proteins in *Nb epss* leaf zones using *Agrobacteria* and, after 48 h, infiltrated *Pf*0-1 *XopQ* bacteria to activate TNL^Roq1 immunity (Fig. 4a). Co-expression of *Arabidopsis* EDS1-FLAG, SAG101-FLAG and NRG1.1-SH led to XopQ-dependent host cell death quantified in electrolyte leakage assays at 24 hpi (Fig. 4b). In this ETI assay, replacement of *Arabidopsis* NRG1.1-SH by ADR1-L2-SH or co-expression of GUS-FLAG with NRG1.1-SH did not lead to XopQ-triggered host cell death (Fig. 4b), consistent with

PAD4 being dispensable for TNL immune responses in *N. benthamiana*[26,29], even with a *Pf*0-1 stimulus.

We performed a time course to monitor the accumulation and associations of NRG1.1-SH with EDS1-FLAG and SAG101-FLAG in *Nb epss* leaf samples harvested at 4, 8, 12 and 24 h after *Pf*0-1 *XopQ* inoculation. Although protein inputs were similar in all samples, a clear NRG1.1–SAG101 and NRG1.1–EDS1 interaction was detected in α-HA immunopurified samples only at 8 and 12 hpi with *Pf*0-1 *XopQ* (Fig. 4c). The immunoprecipitation (IP) signals were no longer detectable at 24 hpi, suggesting that Roq1

recognition of $Pf$0-1 $XopQ$-induced NRG1 association with EDS1 and SAG101 is transient and/or disrupted by host cell death at 24 hpi (Fig. 4b, c). As in *Arabidopsis*, no association between NRG1.1 and EDS1 or SAG101 was detected at 12 hpi with $Pf$0-1 alone (Fig. 4c), further indicating that a PTI response in *N. benthamiana* is insufficient to induce NRG1 association with EDS1 and SAG101. Similarly, when *Arabidopsis* ADR1-L2-SH was expressed instead of NRG1.1-SH in the *Nb epss* TNL$^{Roq1}$ ETI assay, it did not interact with SAG101-FLAG or EDS1-FLAG at 12 hpi (Fig. 4c), mirroring the failure of ADR1-L2 to signal Roq1-triggered host cell death or bacterial resistance in *N. benthamiana* (Fig. 4b)[26]. These data show that XopQ activation of TNL$^{Roq1}$ in *Nb epss* leaves is necessary to induce NRG1.1 interaction with SAG101 and EDS1 proteins from *Arabidopsis*. The similarity between *Arabidopsis* EDS1 family protein associations with RNLs observed in native *Arabidopsis* (Fig. 3) and non-native *N. benthamiana* (Fig. 4) suggests that interaction specificity determines function of the *Arabidopsis* EDS1-SAG101-NRG1s module in TNL immunity. Analyses in both systems also show that a PTI stimulus alone does not promote NRG1 association with EDS1 and SAG101.

**EDS1 EP domain-dependent NRG1 association with EDS1 and SAG101.** Assembly of *Arabidopsis* EDS1 heterodimers with PAD4 or SAG101 is mediated by a short N-terminal EDS1 α-helix (αH) fitting into an N-terminal hydrophobic groove of either partner[30,32,46]. Protein structure–function studies of *Arabidopsis* and tomato EDS1–SAG101 complexes showed that the heterodimer brings into close proximity two α-helical coils (EDS1 αP and SAG101 αN) on the partner C-terminal 'EP' domains, which are essential for TNL ETI signalling[26,29,30,32]. *Arabidopsis* EDS1 residues F419 and H476 are positioned close to SAG101 αN in the dimer cavity (Fig. 5a)[26]. In earlier *Agrobacteria*-only based *Nb epss* reconstitution assays, an *Arabidopsis* EDS1$^{F419E}$ mutation disabled TNL$^{Roq1}$ signalling without disrupting the EDS1-SAG101 heterodimer[26]. In the *Agrobacteria* plus *Pf*0-1 *XopQ* TNL$^{Roq1}$ assay (Fig. 4a), *Arabidopsis* EDS1$^{F419E}$-FLAG and EDS1$^{H476Y}$-FLAG single amino acid exchange variants failed to mediate XopQ/Roq1-dependent host leaf cell death at 24 hpi or *Xanthomonas euvesicatoria* (formerly, *Xanthomonas campestris* pv. *vesicatoria*, *Xcv*) growth at 6 dpi (Fig. 5b, c). Also, substituting NRG1.1-SH with ADR1-L2-SH did not confer TNL$^{Roq1}$-triggered host cell death and *Xcv* resistance (Fig. 5b, c). We monitored the expression of FLAG-tagged EDS1 family proteins by immunoblotting and performed NRG1.1-SH α-HA IP assays on *Pf*0-1 *XopQ*-triggered leaf protein extracts at 10–12 hpi when TNL-induced NRG1.1–EDS1 and NRG1.1–SAG101 associations were strongest (Fig. 4c). As negative controls, PAD4-FLAG was substituted for SAG101-FLAG, GUS-FLAG for EDS1-FLAG, and ADR-L2-SH for NRG1.1-SH (Fig. 5d). While all proteins were detected in input samples, NRG1.1 was detected only when NRG1.1-SH was co-expressed together with functional wild-type EDS1-FLAG and SAG101-FLAG (Fig. 5d). The failure of NRG1.1 to IP SAG101 with GUS replacing EDS1 (Fig. 5d), indicates that NRG1.1 associates specifically with the EDS1–SAG101 heterodimer and not individual EDS1 or SAG101 monomers, upon triggering of TNL$^{Roq1}$ ETI. The strong reduction in NRG1.1 association with EDS1 EP domain non-functional variants EDS1$^{F419E}$ or EDS1$^{H476Y}$ shows that NRG1 fails to associate with a signalling-inactive EDS1–SAG101 heterodimer after Roq1 activation.

**EDS1 and SAG101 signalling requires an intact NRG1 P-loop and N-terminus.** We next examined NRG1 molecular features that might influence its TNL signalling function and association

with EDS1–SAG101. Because RNLs have domain architectures similar to sensor CNL proteins[17], we modelled NRG1.1 onto the cryo-EM structure of the activated *Arabidopsis* CNL receptor ZAR1 which forms a pentameric resistosome[7]. The ZAR1 signalling active pentamer has five exposed N-terminal α-helices (α1) preceding the CC domains of the NLR protomers, which assemble into a potential membrane-associated pore or channel[7]. Structural modelling of *Arabidopsis* NRG1.1 identified two negatively charged N-terminal glutamic acid (E) residues (E14 and E27) (Fig. 6a) positioned similarly to ZAR1 α1-helix residues E11 and E18 that are part of the ZAR1 inner funnel and are necessary for ZAR1 resistosome activity[7]. Two other NRG1.1 N-terminal residues, leucine 21 (L21) and lysine 22 (K22) (Fig. 6a), aligned with L10 and L14 in the ZAR1 α1-helix which promoted ZAR1 membrane association and resistosome signalling[7]. Neither set of ZAR1 α1-helix amino acids was required for effector-induced pentamer assembly[7]. We further identified in *Arabidopsis* NRG1 RNLs the conserved NLR nucleotide-binding domain (NBD) P-loop (GxxxxGK(T/S)) motif (G$^{199}$K$^{200}$T$^{201}$ in *Arabidopsis At*NRG1.1; Fig. 6a; Supplementary Fig. 11), which mediates nucleotide binding[52,53], and induced self-association and/or signalling functions of several sensor NLRs[54,55]. The P-loop was reported to be dispensable for ADR1-L2 conferred resistance during ETI and basal immunity and NRG1.1-conditioned TNL *chs3-2D* auto-immunity[22,25], but was required for ADR1-L2-conditioned *lsd1*-mediated cell death and *NRG1*-dependent TNL$^{Roq1}$ immunity[35,41].

We made amino acid exchanges to alanines in the NRG1.1 predicted α1-helix E11/E27 (denoted EE) and L21/K22 (LK) pairs and a non-functional NRG1.1 G$^{199}$K$^{200}$T$^{201}$/AAA P-loop variant (denoted GKT). When co-expressed with *Arabidopsis* EDS1 and SAG101 in the *Nb epss* TNL$^{Roq1}$ assay, NRG1.1$^{EE}$ and NRG1.1$^{GKT}$ variants were non-functional and NRG1.1$^{LK}$ was partially functional in eliciting host cell death (Fig. 6b). All mutant NRG1.1-HA variants failed to confer *Xcv* resistance (Fig. 6c). The NRG1.1-HA variants were detected on immunoblots, as were co-expressed EDS1-FLAG and SAG101-FLAG proteins (Fig. 6d; left panel). In NRG1.1 α-HA IP assays, the NRG1.1-HA N-terminal EE and LK mutants immunoprecipitated EDS1-FLAG and SAG101-FLAG as efficiently as wild-type NRG1.1-SH (Fig. 6d; right panel). By contrast, the NRG1.1 P-loop GKT mutant displayed a much weaker association with EDS1 and SAG101 (Fig. 6d; right panel). Failure of NRG1.1$^{GKT}$ to interact with EDS1 and SAG101 indicates a requirement for ADP/ATP binding and/or nucleotide exchange at the NRG1.1 nucleotide-binding domain for TNL-induced association with EDS1–SAG101 and immunity. Retention of NRG1.1$^{EE}$ and NRG1.1$^{LK}$ TNL induced association with EDS1–SAG101, but their loss of immunity activity, suggests that an intact NRG1.1 N-terminal putative α-helix is required for NRG1 signalling in TNL ETI as part of or after its TNL-induced association with EDS1–SAG101.

## Discussion

Plant intracellular NLR receptors, activated directly or indirectly by pathogen effectors, provide a critical surveillance mechanism against disease. Activated forms of the two major sensor NLR classes, TNLs and CNLs, assemble into oligomers that are required for immunity signalling and broadly resemble mammalian NLR inflammasome scaffolds[7–9,56]. Two phylogenetically related groups of HeLo domain helper NLRs (NRG1s and ADR1s) and the EDS1 family of plant-specific lipase-like proteins (EDS1, PAD4 and SAG101) mediate signalling downstream of sensor NLRs, leading to transcriptional defences and localised host cell death[19,25,26,28,35].

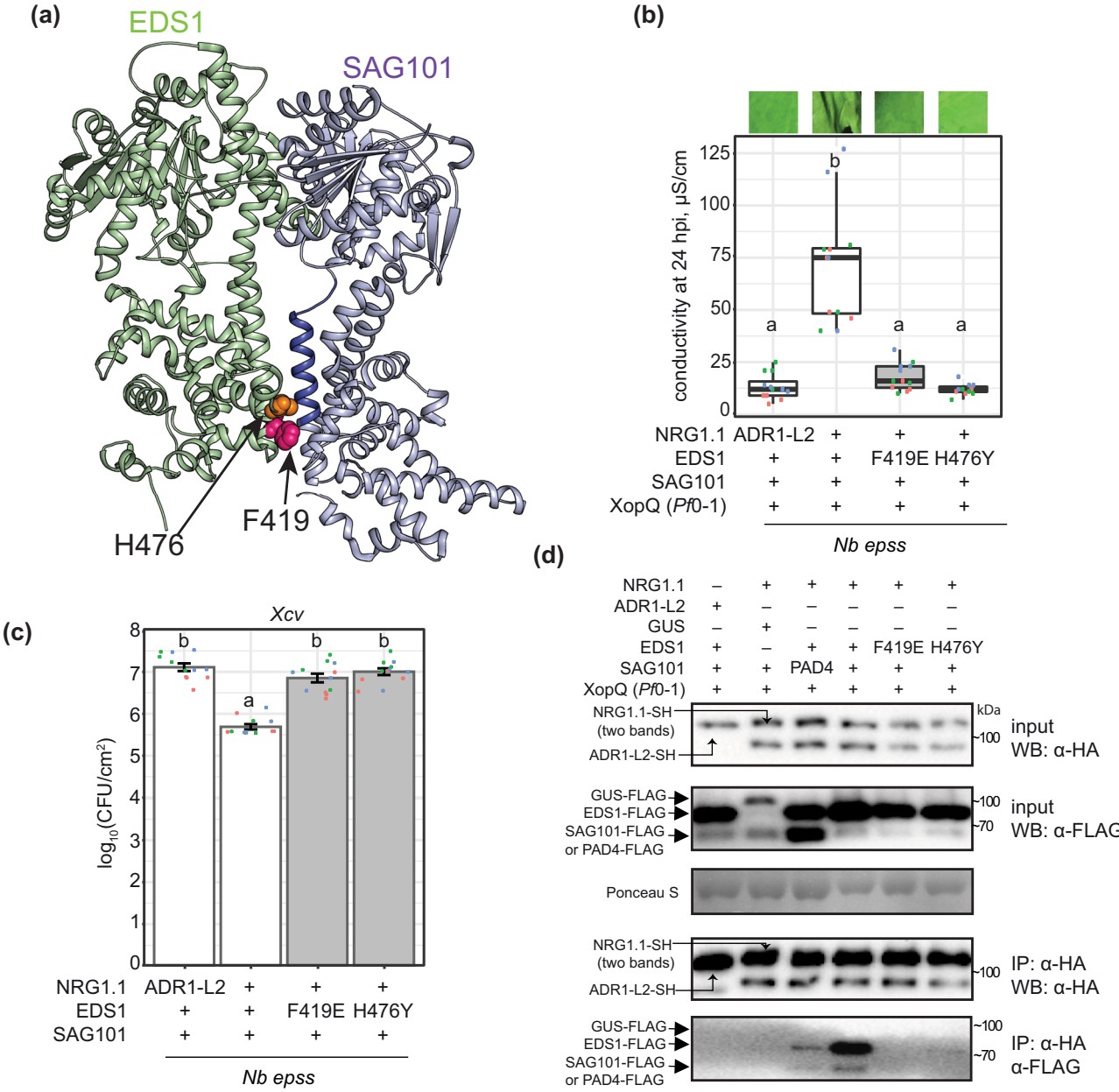

**Fig. 5 Effector-dependent NRG1–EDS1–SAG101 association requires a functional EDS1 EP domain. a** The *Arabidopsis* EDS1 (green)—SAG101 (purple) heterodimer crystal structure (PDB: 4NFU) with highlighted EP domain cavity essential for TNL-triggered cell death. An α-helix of the SAG101 EP domain (blue) and residues F419 (magenta) and H476 (orange) in EDS1 EP domain are shown as ribbon and spheres, respectively. **b, c** Roq1/XopQ-dependent cell death (**b**) and bacterial *Xcv* resistance (**c**) in leaves of *Nb epss* transiently expressing of C-terminally FLAG-tagged EDS1 wildtype and mutant variants with SAG101-FLAG and NRG1.1-SH or ADR1-L2-SH. Cell death was triggered by infiltrating *Pf*0-1 *XopQ* at 48 h after *Agrobacteria* infiltration; photos of leaves were taken at 24 h. F419E and H476Y mutations in the *Arabidopsis* EDS1 EP domain abrogated cell death and resistance. Experiments were performed three times independently, each with four replicates (leaf discs) (Tukey's HSD, $\alpha = 0.001$, $n = 12$). Error bars represent standard error of mean. Description of boxplots: minima—first quartile, maxima—third quartile, centre—median, whiskers extend to the minimum and maximum values but not further than 1.5 inter-quartile range from respective minima or maxima of the boxplot. Datapoints with the same colour come from one independent experiment. **d** IP followed by Western blot analysis to test the dependency of associations between *Arabidopsis* EDS1, SAG101 and NRG1.1 in *Nb epss* on a functional EDS1 EP domain cavity. Leaves of *Nb epss* were infiltrated with *Agrobacteria* to express FLAG-tagged EDS1 or its variants EDS1$^{F419E}$ and EDS1$^{H476Y}$, SAG101-FLAG and NRG1.1-SH or PAD4-FLAG, with ADR1-L2-SH and GUS-FLAG as negative controls. At 2 dpi, *Pf*0-1 *XopQ* (OD$_{600}$ = 0.3) was infiltrated, and the triggered samples were collected at 10 hpi. Following IP with α-HA agarose beads, input and IP fractions were probed with α-HA and α-FLAG antibodies. Roq1/XopQ-dependent association of NRG1.1-SH with EDS1-FLAG and SAG101-FLAG was abolished in samples with mutated EDS1 EP domain variants. Similar results were obtained in three independent experiments. *Xcv Xanthomonas campestris* pv. *vesicatoria* strain 85-10, XopQ (*Pf*0-1) effector tester strain of *Pseudomonas fluorescens* 0–1 delivering XopQ, WB Western blotting, IP immunoprecipitation, *Nb epss Nicotiana benthamiana eds1a pad4 sag101a sag101b*.

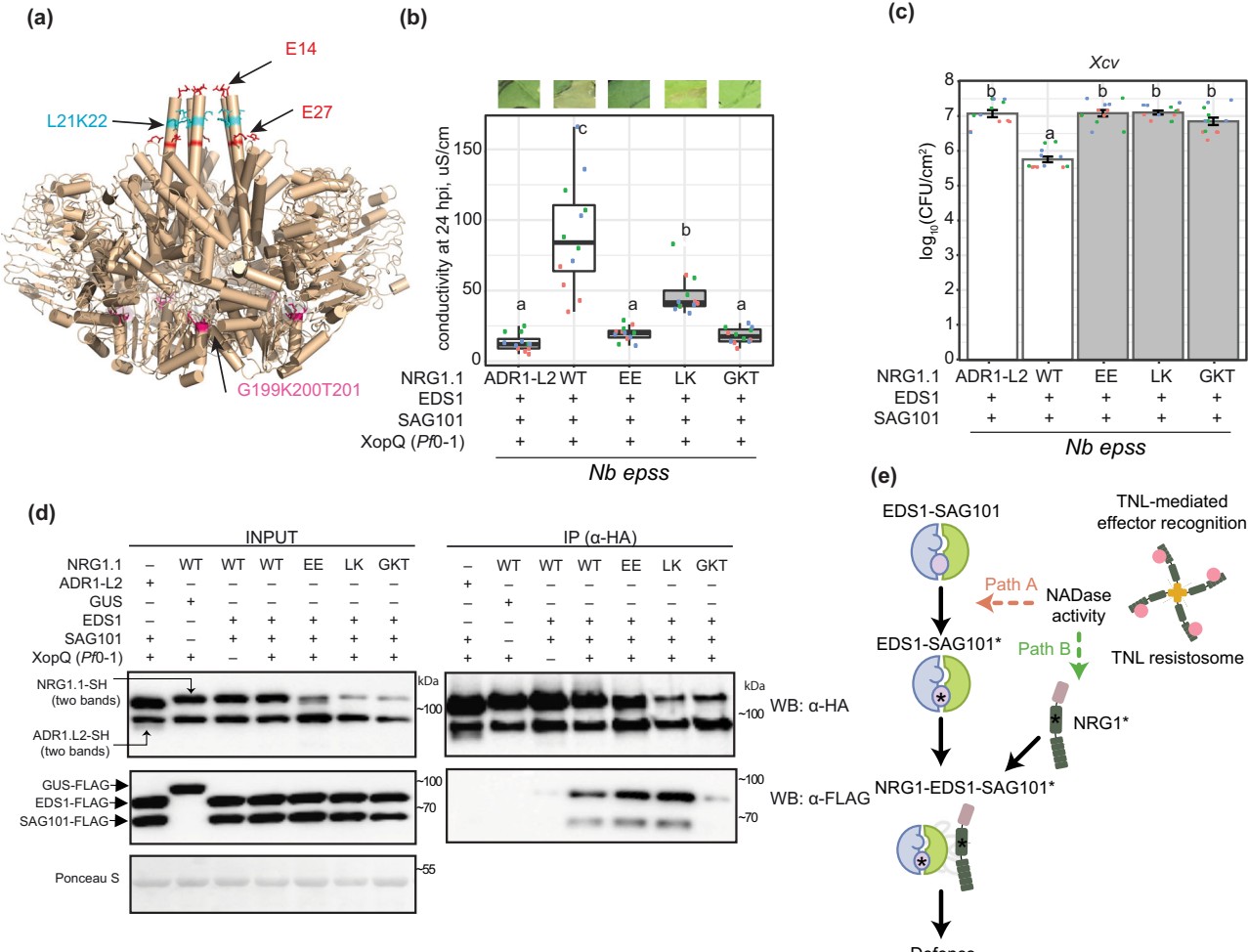

**Fig. 6 Association between *Arabidopsis* NRG1.1, EDS1 and SAG101 require an intact NRG1.1 P-loop. a** A structure homology model of *Arabidopsis* NRG1.1 based on the ZAR1 resistosome. NRG1.1 amino acids E14 and E27 (red sticks) and L21 and K22 (blue sticks) aligned with residues E11, E18, F9, L10 and L14 in ZAR1. The modelled NRG1.1 P-loop motif (G199 K200 T201) is shown as cyan sticks. XopQ-triggered cell death (**b**) and *Xanthomonas campestris* pv. *vesicatoria* strain 85-10 (*Xcv*) growth restriction (**c**) in *Nicotiana benthamiana eds1a pad4 sag101a sag101b* (*Nb epss*) expressing *Arabidopsis* NRG1.1-SH wildtype, NRG1.1$^{E14A/E27A}$, NRG1.1$^{L21A/K22A}$ and NRG1.1$^{G199A/K200A/T201A}$ variants together with EDS1-FLAG and SAG101-FLAG. *Arabidopsis* proteins were expressed via *Agrobacteria*-mediated transient expression assays two days before infiltration of *Pseudomonas fluorescens* 0-1 (*Pf*0-1) *XopQ* (OD$_{600}$ = 0.3) or simultaneously with *Xcv* (OD$_{600}$ = 0.0005). Experiments were performed three times independently, each with four replicates (leaf discs) (Tukey's HSD, $\alpha = 0.001$, $n = 12$). Error bars represent standard error of mean. Datapoints with the same colour come from one independent experiment. **d** Immunoprecipitation (IP) and Western blot (WB) analyses testing associations of *Arabidopsis* NRG1.1 mutant variants with EDS1 and SAG101 in *Nb epss* after triggering Roq1 signalling by *Pf*0-1 *XopQ* infiltration (OD$_{600}$ = 0.3; 10 hpi). *Arabidopsis* EDS1-FLAG, SAG101-FLAG with NRG1.1-SH, and using GUS-FLAG and *Arabidopsis* ADR1-L2-SH as negative controls, were expressed using *Agrobacteria* 2 d prior to *Pf*0-1 *XopQ* infiltration. After α-HA IP, the indicated fractions were analysed with α-HA and α-FLAG antibodies by WB. The experiment was conducted three times independently with similar results. **e** Model of molecular events leading to generation of an *Arabidopsis* EDS1–SAG101–NRG1 signalling complex essential for TNL receptor-activated defence. NRG1–EDS1–SAG101 association is dependent on TNL activation and requires an intact EDS1–SAG101 heterodimer EP domain cavity (purple circle) and NRG1 nucleotide-binding domain. In two depicted scenarios, an effector-induced TNL oligomer with NADase activity leads to activation (asterisks) of EDS1–SAG101 via the EP domain cavity ('Path A', asterisk inside the purple circle) or NRG1 ('Path B'). These paths are not mutually exclusive. EDS1–SAG101–NRG1 assembly precedes and is necessary for TNL-triggered cell death and resistance involving a predicted NRG1 N-terminal HeLo domain α-helix.

Here we provide genetic and molecular evidence in *Arabidopsis* that distinct RNL immunity modules (or branches) operate with specific EDS1 family heterodimers. The two modules contribute in different ways to TNL$^{RRS1–RPS4}$ and CNL$^{RPS2}$ ETI, and to basal immunity against virulent bacteria (Figs. 1 and 2; Supplementary Figs. 1–6). We show in *Arabidopsis* and *N. benthamiana* that TNL-effector activation induces a specific association between NRG1 proteins and EDS1–SAG101 heterodimers (Figs. 3 and 4). In *N. benthamiana* TNL$^{Roq1}$ ETI reconstitution assays, interactions of NRG1 with EDS1 and SAG101 and effective TNL immunity signalling require a functional EDS1 EP domain within the EDS1–SAG101 heterodimer and an intact P-loop motif in NRG1 for nucleotide binding (Figs. 5 and 6). Conserved amino acids at the NRG1 N-terminus, modelled onto the structure of a CNL receptor (ZAR1) membrane pore-forming α1-helix[7], are important for TNL ETI but not for TNL-induced NRG1 association with EDS1–SAG101 (Fig. 6). Our data provide a molecular underpinning for genetically separate RNL-EDS1 mechanisms conferring immunity and cell death downstream of NLR-mediated effector recognition (Figs. 1 and 2)[25,26,28]. The

data also suggest that sensor NLR-induced assembly of helper NLRs with EDS1 family proteins is a critical step in TNL downstream signalling.

Previous studies in *Arabidopsis* revealed unequal genetic and transcriptional contributions (unequal redundancy) of *PAD4* vs. *SAG101*[19,25,26,31] and *ADR1* vs. *NRG1* groups in ETI and basal immunity responses to pathogens[12,26]. *Arabidopsis SAG101* and *NRG1*s function in ETI mediated by TNL receptors and are drivers of TNL host cell death and transcriptional reprogramming[12,26,27]. *PAD4* and *ADR1*s are recruited more broadly for TNL and CNL ETI immune responses, in which they control transcriptional SA-dependent and SA-independent defence pathways[22,28,32,33]. Our analysis of *Arabidopsis pad4, sag101, adr1-* and *nrg1*-group combinatorial mutants shows that individual components of the two immunity modules are not interchangeable in TNL$^{RRS1-RPS4}$ or CNL$^{RPS2}$ immune responses. These data reveal a specificity in module composition and function (Fig. 1; Supplementary Figs. 1–3).

Removing the *ICS1*-dependent SA pathway uncovered the extent to which separate *PAD4-ADR1*s and *SAG101-NRG1*s genetic mechanisms are preserved (Fig. 2; Supplementary Figs. 5 and 6). The SA pathway is bolstered in *Arabidopsis* by *PAD4* and *ADR1*s via a mutually reinforcing feedback loop[22,33,41,57]. Notably, the removal of *ICS1* and *SAG101* together, released a *PAD4-ADR1*s activity leading to host cell death in TNL$^{RRS1-RPS4}$ immunity (Fig. 2b, c). Hence, each TNL signalling branch has a transcriptional reprogramming and cell death-inducing capacity, depending on the status of other pathways in the network[5,58,59]. *SAG101-NRG1*s and SA antagonism of *PAD4-ADR1*s-stimulated cell death suggests there is crosstalk between the different sectors, possibly to limit host tissue damage and promote systemic relay of immunity signals[28]. This aligns with a reported role of SA receptor non-expressor of PR1 (NPR1) in limiting NLR cell death in the presence of high SA concentrations around the tissue undergoing cell death[60]. Although metacaspase 1 (MC1) controlled proteolysis promoting RPM1 cell death is also conditionally antagonised by SA[39], we did not detect a role for *MC1* in *SAG101-NRG1*s or *PAD4-ADR1*s stimulated TNL$^{RRS1-RPS4}$ triggered cell death (Supplementary Fig. 7).

*Arabidopsis* IP–LC–MS analyses using EDS1, SAG101, PAD4 and NRG1.2 individually as baits were performed in TNL$^{RRS1-RPS4}$ triggered leaf tissues between 4 and 8 h after bacterial infiltration, based on knowledge of a critical 4–8 h time window needed for EDS1-PAD4 and ADR1s mobilised gene expression to be effective in immunity[12,13,32,43,49]. Also, *Arabidopsis* TNL$^{RRS1-RPS4}$ triggered *SAG101-NRG1*s-dependent cell death started to increase from 6–8 hpi with *Pf*0-1 *avrRps4* (Fig. 2; Supplementary Fig. 1)[26]. The sum of IP–MS data (Fig. 3) points to molecular specificity in EDS1–SAG101 interaction with NRG1.1 and NRG1.2 functional isoforms, and in EDS1–PAD4 interaction with ADR1-L1 and ADR-L2 isoforms, in TNL receptor-induced cells. The dynamics of EDS1–SAG101–NRG1s associations relative to presumed EDS1–PAD4–ADR1s associations in immune-activated cells and tissues remain unresolved. The absence of a detectable association between NRG1.2 with EDS1 or SAG101 at 4 h after *Pf*0-1 EV treatment (Fig. 3d), shows that PTI alone is insufficient to induce an NRG1 interaction with EDS1 or SAG101. These data imply that NRG1.2 association with EDS1 and SAG101 requires an activated TNL-derived signal. We suggest it is likely that an EDS1–SAG101–NRG1 functional interaction is principally a post-transcriptional event because (i) it was detected in *Arabidopsis* at 4 hpi (Fig. 3d, e) before the main *EDS1*-dependent transcriptional elevation at 8–10 hpi[12,32,49] and (ii) it could be recapitulated in *N. benthamiana* with abundant transiently expressed proteins only after an effector-TNL trigger (Fig. 4c). Although we detected a specific association between

PAD4 and ADR1s in TNL immune-activated *Arabidopsis* tissues (Fig. 3; Supplementary Fig. 9; Supplementary Data 2), it has yet to be determined whether their interaction is dependent on a PTI and/or ETI trigger. The preferential enrichment of several cell surface PTI-related proteins by PAD4 in TNL-triggered *Arabidopsis* tissues (Supplementary Fig. 9) is consistent with the EDS1-PAD4-ADR1 node contributing to proposed connectivity between ETI and PTI responses[15,16,61].

Interestingly, signalling inactive truncated NRG1.3 isoform[25] was enriched with SAG101 and to a lesser extent with PAD4 (Fig. 3a; Supplementary Fig. 9b). This might reflect an NRG1.3 role in the negative regulation of both modules. Indeed, *Arabidopsis* NRG1.3 appears to interfere with the defence signalling by NRG1.1 and NRG1.2[62]. It is also possible that NRG1 determinants for preferential association with EDS1–SAG101 lie in the NRG1 C-terminal portion. Weak association detected between PAD4 and NRG1.1 (Fig. 5d) also deviates from otherwise clear-cut specific interactions of within-module components (Fig. 3a). However, these associations are unlikely to contribute to *Arabidopsis* TNL immune signalling (Fig. 1)[26].

We interrogated the molecular requirements for *Arabidopsis* EDS1 and SAG101 functional association with NRG1.1 in *Nb epss* TNL$^{Roq1}$ transient reconstitution assays. TNL ETI-induced NRG1.1 association was only observed in IPs with EDS1 and SAG101 together (Figs. 4c, 5d, 6d), supporting NRG1 interaction with a signalling competent EDS1–SAG101 heterodimer (Fig. 3) but not with EDS1 or SAG101 individually (Fig. 5d) which are inactive[30,45,46]. A previous study proposed that NRG1 signals downstream of EDS1 in regulating TNL$^{Roq1}$ immunity and cell death[35]. Our protein interaction assays instead point to NRG1 working biochemically together with EDS1–SAG101 in the TNL$^{Roq1}$ immunity signalling cascade, although we cannot exclude NRG1 subsequently dissociating from an EDS1–SAG101 complex as part of the immune response. The discrete timing of NRG1–EDS1–SAG101 interaction detection between 8 and 12 hpi suggests it is transient in nature, although it is unclear in this system whether reduced association at later time-points is a controlled event, possibly to dampen outputs, or due to cell death.

A requirement for EDS1 EP-domain essential residues within the EDS1–SAG101 heterodimer[26,29] to interact with NRG1.1 in a TNL ETI-dependent manner (Fig. 5d), suggests that an intact EDS1–SAG101 EP-domain drives its association with NRG1 downstream of TNL receptor activation, as depicted in a model (Fig. 6e). TNL-induced NRG1–EDS1–SAG101 association and ETI also required a nucleotide-binding form of NRG1.1, whereas N-terminal NRG1.1 amino acids on a ZAR1-like functional N-terminal α-helix, were dispensable for their association (Fig. 6d). Together, these data argue for TNL effector recognition rendering both NRG1 and EDS1–SAG101 competent to interact and confer pathogen resistance (Fig. 6e). Recently, the TIR domains of plant TNLs and certain truncated TIR forms were shown to exhibit an NADase activity (shown in Fig. 6e) that is necessary to induce plant *EDS1*-dependent cell death[10,11,63]. Reported cryo-EM structures of effector-activated TNLs *Arabidopsis* RPP1 and *N. benthamiana* Roq1 reveal them to be tetrameric complexes with imposed TIR domain orientations creating an active NADase enzyme[8,9]. Our findings suggest concerted actions between plant TNL NADase activity and immunity signalling via TNL-effector recognition induced EDS1–SAG101–NRG1 association. In our model (Fig. 6e), we envisage two paths for generating a signalling competent NRG1–EDS1–SAG101 complex. In path A, a TNL-derived NADase product binds to EDS1–SAG101, thereby enabling EP domain-dependent association with NRG1, perhaps promoting NRG1 oligomerization. In path B, the activated TNL receptor and/or NADase products cause an NRG1 nucleotide-dependent conformational change

(independently of EDS1–SAG101) which promotes its association with EDS1–SAG101. The data presented here represent a significant advance by showing that pathogen-activated TNL receptors mediate downstream signalling via induced, specific interactions between RNLs and EDS1 family proteins.

## Methods

**Plant materials and growth conditions.** *Arabidopsis thaliana* L. Heynh. (hereafter *Arabidopsis*) wild type, transgenic and mutant lines as well as *N. benthamiana* lines are listed in Supplementary Table 1. For pathogen growth and cell death assays, *Arabidopsis* plants were grown under short-day conditions (10 h light 22 °C/14 h dark 20 °C, light intensity of ~100 μmol m$^{-2}$ s$^{-1}$, 65% relative humidity) for 4–5 weeks. Crosses and seed propagation were conducted under speed breeding growth conditions: 22 h light 22 °C/2 h dark 20 °C, ~100 μmol m$^{-2}$ s$^{-1}$, 65% relative humidity. *N. benthamiana* plants were grown in a greenhouse under long-day conditions for 5–6 weeks.

**Cloning and generation of complementation lines.** Genomic Col-0 *SAG101* sequence (AT5G14930.2) including the coding and upstream (−992 bp) sequences were cloned into pENTR/D-TOPO (K240020, Thermo Fisher Scientific) and further LR-inserted (11791100, Thermo Fisher Scientific) into the expression vector pXCG-mYFP[63] resulting in pXCG pSAG101:SAG101-YFP. pXCSG p35S:NRG1.1-StrepII-3xHA as well as the pXCSG pADR1-L2:ADR1-L2-StrepII-3xHA constructs were described previously[26]. Constructs to express NRG1.1 mutant variants were prepared using the Golden gate MolClo kit[64]. The genomic sequence of Col-0 *NRG1.1* (AT5G66900.1, from start ATG codon to the last codon position) was cloned into the level 0 plasmid pAGM1287, and the *NRG1.1* variants were generated following the QuikChange II Site-Directed mutagenesis (SDM) protocol (#200555, Agilent). Level 0 golden gate compatible construct for the genomic sequence of Col-0 *EDS1* (AT3G48090.1, from the first to the last codon) was synthesised and inserted into the pAM vector (GeneArt, Thermo Fisher Scientific). *EDS1*[H476Y] and *EDS1*[F419E] mutant constructs were generated via SDM. Primers for cloning and SDM are listed in Supplementary Data 1. To obtain level 1 expression constructs, level 0 constructs of *NRG1.1* mutants were combined with the cauliflower mosaic virus (CaMV) *35S* promoter (pICH51288), hemagglutinin tag (3xHA, pICSL50009), CaMV *35S* terminator (pICH41414) and the backbone pICH47732. Expression vectors for *EDS1* (p35S:EDS1[H476Y]-3xFLAG, p35S:EDS1[F419E]-3xFLAG) were cloned following the same strategy except the tag module was replaced by 3xFLAG (pICSL50007). The wild type p35S:EDS1-3xFLAG was prepared by LR-recombining pENTR/D-TOPO EDS1_noStop (genomic sequence of AT3G48090.1 from ATG to the last codon[30] with pAMPAT-3xFLAG[64]. p35S:PAD4-3xFLAG expression construct is a result of a LR reaction between pENTR/D-TOPO PAD4[30] and pAMPAT-3xFLAG. p35S:3xFLAG-GUS was prepared by LR-recombining pJ2B-3xFLAG[64] with pENTR GUS (from LR clonase II kit, 11791020, Thermo Fisher Scientific). To prepare the *PAD4* expression construct for complementation, we PCR-amplified the *PAD4* locus (AT3G52430.1) from the upstream gene's stop codon (AT3G52420) up to the start codon of the downstream gene (AT3G52440) and placed it in a pDONR201 vector via PIPE-PCR[65]. Subsequently, N-terminal YFP with a linker peptide (Gly followed by 9x Ala) was introduced via PIPE-PCR as well. This construct was LR-recombined in a pAlligator2 destination vector[66]. Level 1 golden gate and gateway expression constructs were transformed via electroporation into *Rhizobium radiobacter* (former *Agrobacterium tumefaciens* or *Agrobacteria*) GV3101 pMP90RK or pMP90 for transient expression in *N. benthamiana* and stable expression in *Arabidopsis* (Supplementary Table 2). We transformed pXCG pSAG101:SAG101-YFP and pAlligator2 pPAD4:YFP-Linker-PAD4 into *Arabidopsis pad4-1 sag101-3* mutant and selected homozygous complementation lines using BASTA resistance or the GFP seed coat fluorescence markers, respectively.

**Pseudomonas growth and cell death assays in Arabidopsis.** *Pseudomonas syringae* pv. *tomato* (*Pst*) DC3000 with the empty vector pVSP61, pVSP61 avrRps4 or pVSP61 avrRpt2 were syringe-infiltrated into *Arabidopsis* leaves at OD$_{600}$ = 0.0005 in 10 mM MgCl$_2$. Leaf discs were harvested at 0 dpi (four leaf discs as four replicates) and 3 dpi (12 leaf discs divided over four replicates). Biological replicates are experiments performed on different days with the same or different batches of plants. For cell death assays, a type III secretion system-equipped *Pseudomonas fluorescens* effector tester strain *Pf*0-1 avrRps4[48] was resuspended in 10 mM MgCl$_2$ (OD$_{600}$ = 0.2) and syringe-infiltrated into leaves. Only the type III secretion system-equipped and not the wild type *Pf*0-1 strain was used in this study, and for simplicity, we refer to it as "*Pf*0-1". Eight leaves per genotype were infiltrated for each biological replicate (experiments performed on different days with the same or different batches of plants). The conductivity of solution with the incubated leaf discs was measured at 6, 8, 10, and 24 hpi as described earlier[26]. Macroscopic cell death phenotype was recorded at 24 hpi. For cell death assays with *Pst* avrRpt2, bacteria were resuspended 10 mM MgCl$_2$ to OD$_{600}$ = 0.02 and electrolyte leakage was measured as described for *Pf*0-1 avrRps4 triggered cell death. The statistical analysis included checking normality of residuals distribution (Shapiro–Wilcoxon at $\alpha$ = 0.05 or visually with qq-plot) and homoscedasticity (Levene test at $\alpha$ = 0.05).

Difference in means was assessed via Tukey's HSD test ($\alpha$ = 0.001, experiment taken as a factor in ANOVA).

**Roq1 signalling reconstitution assays in N. benthamiana.** *Roq1* cell death reconstitution assays were performed with the *Pf*0-1 XopQ[67] strain in *N. benthamiana* quadruple mutant *eds1a pad4 sag101a sag101b* (*Nb epss*)[26]. Agrobacteria induced for one hour in Agromix (10 mM MgCl$_2$, 10 mM MES pH 5.6, 150 μM acetosyringone) were firstly syringe-infiltrated at OD$_{600}$ = 0.2 into *Nb epss* leaves. At 48 hpi, *Pf*0-1 XopQ or *Pf*0-1 (empty vector) were infiltrated at OD$_{600}$ = 0.3 in the same leaf zone. At 24 h after *Pf*0-1 XopQ infiltration, macroscopic cell death phenotype was recorded and four-leaf discs (as four replicates) were taken for measuring electrolyte leakage at 6 h after collecting the leaf discs. *Xanthomonas campestris* pv. *vesicatoria* (*Xcv* 85-10; also named *Xanthomonas euvesicatoria*) growth assays in *Nb epss* in the presence of Agrobacteria to express proteins of interest were performed as described earlier[26].

**Co-IP and immunoblotting analyses.** In Co-IP assays with proteins expressed in *N. benthamiana Roq1* reconstitution assays, five 10 mm leaf discs were collected to form a sample. Total protein from the plant material ground to fine powder was extracted in 2 ml of the extraction buffer (10% glycerol, 100 mM Tris–HCl pH 7.5, 5 mM MgCl$_2$, 300 mM NaCl, 10 mM DTT, 0.5% NP-40, 2% PVPP, 1x Plant protease cocktail (11873580001, MilliporeSigma)). Lysates were centrifuged for 35 min at 4500 × g and filtered through two layers of Miracloth (475855, MilliporeSigma). The 50 μl aliquots of the filtered supernatant were taken as input samples. Co-IP were conducted for 2 h with 12 μl α-HA affinity matrix (11815016001, MilliporeSigma) under constant rotation. Beads were collected by centrifugation at 4000 × g for 1 min and washed four times in extraction buffer (without DTT and PVPP). All co-IP steps were conducted at 4 °C. Beads and input samples were boiled at 95 °C in 100 μl 2 × Laemmli buffer for 10 min. Antibodies used for immunoblotting were α-GFP (11814460001, MilliporeSigma), α-HA (1:5000; c29f4, Cell Signalling), α-FLAG (1:5000; f1804, MilliporeSigma), HRP-conjugated antibodies (A9044 and A6154, Sigma-Aldrich; sc-2006 and sc-2005, Santa Cruz). Antibodies were used in dilution 1:5000 (TBST with 3% non-fat milk powder). Images of blots are provided in the Source data file accompanying this paper.

**Immunoprecipitation of EDS1-YFP and TRB1-GFP in Arabidopsis.** Nuclei-enriched fractions were isolated from ~20 g of leaves of 4–5 week-old plants grown under short day conditions described above and vacuum-infiltrated with *Pst avrRps4* (OD$_{600}$ = 0.1, 10 mM MgCl$_2$ supplemented with 0.005% Silwet-L77). Samples were collected at 8–10 h after the infiltration. After sample collection, all steps were performed at 4 °C or on ice. Extraction was performed mainly as in ref. [68] with modifications. Leaves were chopped with a razor blade, all buffers contained a 2xPlant Protease inhibitor cocktail (11873580001, MilliporeSigma). Separation was performed only on one Percoll gradient (80–30%) followed by a final clean-up through the 30% Percoll layer. Nuclei-enriched fractions were spun down for 10 min at 1000 × g to remove Percoll and hexylene glycol.

The nuclei-enriched fraction was gently resuspended in 1 ml of the sample buffer (20 mM Tris–HCl pH 7.4, 2 mM MgCl$_2$, 150 mM NaCl, 5% glycerol, 5 mM DTT, EDTA-free protease inhibitor (11873580001, MilliporeSigma)), centrifuged for 10 min at 1000 × g, and carefully resuspended again in 600 μl of the sample buffer. After incubation of washed nuclei at 37 °C with 10 units of DNase I (89836, Thermo Fisher Scientific) and 20 μg of RNase A (EN0531, Thermo Fisher Scientific) under soft agitation for 15 min, samples were placed on ice for 10 min and sonicated for six cycles 15 s "on"–15 s "off" using Bioruptor Plus (Diagenode). After that, samples were centrifuged for 15 min at 16,000 × g, and the supernatant was used as input for IP with GFP-trap A beads (per IP, 25 μl of slurry prewashed in 2 ml of the samples buffer, gta-100 (Proteintech)). Before IP, 25 μl of the supernatant is set aside as an input fraction for quality controls. Then, samples were supplemented with 10% Triton X-100 to the final concentration 0.1% and 0.5 M EDTA to the final concentration of 2 mM. After 2.5 h of incubation with the sample in Protein Lobind tubes (0030108116 and 0030108132, Eppendorf), beads were washed four times in 300 μl 3 min each in the wash buffer (20 mM Tris–HCl pH 7.4, 150 mM NaCl, 2 mM EDTA). Proteins were eluted in 2 × 35 μl 0.1% TFA and neutralised in 90 μl of Tris–Urea (4 M urea 50 mM Tris–HCl pH 8.5).

**Gel filtration chromatography.** Nuclear extracts (600 μl) from the *pEDS1:EDS1-YFP* complementation line were processed as for IP input in the subsection "Immunoprecipitation (IP) of EDS1-YFP and TRB1-GFP from respective *Arabidopsis* complementation lines". Obtained samples were fractionated on the column Superose 6 10/300 GL (50 kDa–5 MDa range, GE Healthcare Life Sciences, Äkta FPLC) at the rate 0.5 ml/min in 20 mM Tris–HCl pH 7.4 and 150 mM NaCl. The temperature was kept at 4 °C. In total, 28 0.5 ml fractions per sample were collected, concentrated with StrataClean resin (400714, Agilent) and analysed using Western blot method (α-GFP, 11814460001, MilliporeSigma) with the same total EDS1-YFP sample on each blot for the between-blot normalisation. A high-molecular weight marker was run prior to each experiment (28403842, GE Healthcare Life Sciences).

**Immunoprecipitation of YFP-PAD4 and SAG101-YFP in *Arabidopsis*.** Five-week-old *Arabidopsis* plants containing p35S:StrepII-3xHA-YFP (Col-0), pPAD4: YFP-PAD4 (*pad4-1 sag101-3* background) or pSAG101:SAG101-YFP (*pad4-1 sag101-3*) were vacuum infiltrated with *Pf*0-1 *avrRps4* bacteria (OD = 0.2 in 10 mM MgCl$_2$ with 0.01% Silwet L-70). ~2 g of rosette material was collected at 6 hpi, snap-frozen in liquid nitrogen and kept at −80 °C until IP. On the day of IP, samples were ground to fine powder in Precellys 15 ml tubes (P000946-LYSK0-A, Bertin Instruments). The protein extraction was performed in 10 ml of the buffer composed of 20 mM PIPES–KOH pH 7.0, 150 mM NaCl, 10 mM MgCl$_2$, 10% glycerol (v/v), 5 mM DTT, 1% Triton X-100, Plant Protease Inhibitor cocktail (11873580001, MilliporeSigma). The protein extraction was performed at 4 °C for 20 min under constant end-to-end mixing (~60 rpm). After that, the samples were cleared by centrifuging 20 min at 4 °C 3000 × *g*. The supernatant was passed once through 0.2 μm filters (KC64.1, Roth) to remove debris. Each sample (10 ml in 15 ml Falcon tubes) was incubated for 2.5 h at 4 °C under constant end-to-end mixing (~20 rpm) with equilibrated beads corresponding to 20 μl of GFP-trapA (gta100, Proteintech) slurry. After the incubation, beads were washed three times 5 min each in the wash buffer containing 20 mM Tris–HCl pH 7.4, 150 mM NaCl, 0.01% Triton X-100, Plant Protease Inhibitor cocktail (11873580001, MilliporeSigma). Finally, to remove Triton X-100 traces, the beads were washed two additional times 1 min each in the buffer with 20 mM Tris–HCl pH 7.4 and 150 mM NaCl.

**Purification of NRG1.2 from the *Arabidopsis* complementation line.** *Arabidopsis* Ws-2 *nrg1a nrg1b* complementation line from[19] was grown in short-day conditions for 5–6 weeks. *Pf*0-1 empty vector or *Pf*0-1 *avrRps4* were resuspended in the infiltration buffer (10 mM MES pH 5.7, 10 mM MgCl$_2$) at OD$_{600}$ = 0.3 and syringe-infiltrated into apoplastic space of rosette leaves. Approximately 2.5 g of tissue was harvested 4 hpi and flash frozen. Tissue was ground in liquid nitrogen before lysis in 100 mM HEPES (pH 7.5), 300 mM NaCl, 5 mM MgCl$_2$, 0.5% Nonidet P-40, 10 mM DTT, 10% glycerol, and cOmplete™ EDTA-free Protease Inhibitor Cocktail (11873580001, MilliporeSigma). Lysate was centrifuged at 4000 × *g* for 35 min at 4 °C and filtered through Miracloth to pellet and remove debris. Lysate was incubated with buffer equilibrated ANTI-FLAG® M2 Affinity Agarose beads (A2220, MilliporeSigma) for 45 min. Beads were washed before incubation with 3XFLAG® Peptide (F4799, Sigma-Aldrich) for 2 h. Eluate was collected for further analysis by immunoblot and mass spectroscopy.

**Immunoblot analysis of NRG1.2-EDS1 interactions in *Arabidopsis*.** Samples were heated in 4× SDS sample loading buffer (10 mM DTT) at 65 °C for 5 min. Proteins were resolved on 4–20% SDS–PAGE (4561095, Bio-Rad) and dry transferred by Trans-Blot Turbo Transfer System to PVDF membrane (170427, Bio-Rad). Membranes were blocked in 5% milk (v/w) in TBST for 1 h. For protein detection, HRP-conjugated anti-FLAG (A-8592, Sigma-Aldrich) was used at 1:30,000 (TBST, 5% milk powder [v/w]) and anti-EDS1 (AS13 2751, Agrisera) was used at 1:3000 (TBST, 3% milk powder [v/w]) and probed overnight at 4 °C. Membranes were washed three times in TBS-T for 10 min. Secondary HRP-conjugated antibody (A0545, MilliporeSigma) was used at 1:10,000 (TBST, 5% milk powder [v/w]) at RT for 2 h. Membranes were washed three times in TBST for 10 min, and three times in TBS for 5 min. Detection of signal was performed with enhanced chemiluminescent horseradish peroxidase substrates SuperSignal™ West Pico PLUS (34580, Thermo Fischer Scientific) and Femto (34095, Thermo Fischer Scientific), and ImageQuant LAS 4000™ for protein band visualisation. Images of blots are provided in the Source data file accompanying this paper.

***Arabidopsis* NRG1.1 structure homology modelling.** NRG1.1 (AT5G66900.1) was modelled on ZAR1 resistosome cryo-electron microscopy structure (PDB: 6j5t) using SWISS-MODEL[69] (SMTL version 2019-05-22, PDB release 2019-05-17; ProMod3 v. 1.3.0). Visualisation was performed in Pymol (v. 2.3.4–2.4.1, Schrödinger, LLC).

**Software used for visualisation and statistical analysis of data.** Statistical analysis was performed in R (3.6.1 through 4.0.2) using base ANOVA and posthoc tests. Heatmaps were generated with the pheatmap (1.0.12) package in R. Boxplots and barplots were generated with the ggplot2 (3.3.2) package in R. Multiple sequence alignment in Supplementary Fig. 11 was prepared with msa package (1.22.0) in R. An NRG1.1 structure model was prepared with SWISS-MODEL[69] (SMTL version 2019-05-22, PDB release 2019-05-17; ProMod3 v. 1.3.0). Visualisation of the structures was performed in Pymol (2.3.4-2.4.1, Schrödinger, LLC). Excel in Microsoft Office suites 2016, 2019, 365 were used for the analysis of processed mass spectrometry data.

**Reporting summary.** Further information on research design is available in the Nature Research Reporting Summary linked to this article.

## Data availability
The mass spectrometry data generated in this study have been deposited in the PRoteomics IDEntification (PRIDE) database under accession codes: P2; P1; P7. The sequence data analysed for the NLRs in the alignment in Supplementary Fig. 11 are NRG1.1–AT5G66 900.1 (TAIR, [https://www.arabidopsis.org/servlets/TairObject?type=locus&name=At5g 66900]); NRG1.2–AT5G66910.1 (TAIR, [https://www.arabidopsis.org/servlets/TairObject? type=locus&name=At5g66910]); *Lus*NRG1-Lus10022464 (Phytozome, [https://phytozome. jgi.doe.gov/pz/portal.html#!gene?search=1&detail=1&method=3127&searchText=transcri ptid:23171720]); *Nb*NRG1-Niben101Scf02118g00018.1 (SolGenomics, [https://solgenomics. net/jbrowse_solgenomics/?data=data%2Fjson%2FNiben1.0.1&loc=Niben101Scf02118% 3A107051..119466&tracks=DNA%2CNibenv101_gene_models&highlight=]); *At*ADR1-L2-AT5G04720.1 (TAIR, [https://www.arabidopsis.org/servlets/TairObject?type=locus& name= At5g04720]), *Sl*ADR1-Solyc04g079420.3.1 (SolGenomics, [https://solgenomics.net/ locus/110992/view]); *Nb*ADR1-Niben101Scf02422g02015.1 (SolGenomics, [https://solgeno mics.net/jbrowse_solgenomics/?data=data%2Fjson%2FNiben1.0.1&loc=Niben101Scf 02422%3A298959..305123&tracks=DNA%2CNibenv101_gene_models&highlight=]); *Sl*NRC4-Solyc04g007070.3.1 (SolGenomics, [https://solgenomics.net/locus/20361/view]); *Sl*NRC3-XP_004238948.1 (NCBI, [https://www.ncbi.nlm.nih.gov/protein/XP_004238948.1/ ]); *At*ZAR1-AT3G50950.2 (TAIR, [https://www.arabidopsis.org/servlets/TairObject?type= locus&name= AT3G50950]); N-Q40392 (Uniprot, [https://www.uniprot.org/uniprot/ Q40392]); Roq1-ATD14363.1 (NCBI, [https://www.ncbi.nlm.nih.gov/protein/ATD143 63.1]); RPP4 - F4JNA9 (Uniprot, [https://www.uniprot.org/uniprot/F4JNA9]); RPS4-Q9XGM3 (Uniprot, [https://www.uniprot.org/uniprot/Q9XGM3]); RPM1-Q39214 (Uniprot, [https://www.uniprot.org/uniprot/Q39214]); Rx-Q9XGF5 (Uniprot, [https://www. uniprot.org/uniprot/Q9XGF5]). The CryoEM structure of *At*ZAR1 analysed and used as a template-6j5t (PDB [https://www.rcsb.org/structure/6J5T]); EDS1-SAG101 X-ray structure–4nfu (PDB, [https://www.rcsb.org/structure/4NFU]). All other data are provided in the article and its Supplementary files or from the corresponding author upon reasonable request. Source data are provided with this paper.

## Code availability
Custom scripts for statistical analysis described in GitHub repository for Lapin et al. (2019)[26] are also available at Zenodo [https://zenodo.org/record/4660032#. YGcSAy0Rq_w]

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

## Acknowledgements

We thank Franziska Turck (MPIPZ) for providing seeds of the TRB1-GFP complementation line and Deepak D. Bhandari for help with gel filtration chromatography. This work was supported by the Max-Planck Society and Deutsche Forschungsgemeinschaft (DFG) grants SFB 680 (J.E.P., D.L.) and SFB-1403–414786233 (J.E.P., X.S.); DFG-ANR

Trilateral ("RADAR" grant to J.E.P. and J.A.D.) and a Chinese Scholarship Council doctoral fellowship to X.S.

## Author contributions

X.S., D.L., J.M.F., H.N., I.F., J.D.G.J., J.E.P. designed experiments and analysed data; X.S., J.A.D., J.B., J.R., S.B. generated and characterised genetic material; X.S. and J.R. performed cell death and pathogen growth assays; D.L., J.M.F., K.K., S.C.S., A.H., P.D., F.L.H.M, I.F., H.N. contributed immunoprecipitation and LC-MS data; X.S. designed and performed protein-protein interaction assays in *N. benthamiana*. X.S., D.L., J.M.F., J.D.G.J., J.E.P. wrote the manuscript with contributions from all authors.

## Funding

## Competing interests

The authors declare no competing interests.
