## [Peer Review File · Nature Communications]

REVIEWER COMMENTS

Reviewer #1 (Remarks to the Author):

The manuscript by Sun and co-workers entitled "Pathogen effector recognition-dependent association of NRG1 with EDS1 and SAG101 in TNL receptor immunity" describes how two groups of helper NLRs in Arabidopsis form distinct complexes with specific EDS1 family members to appropriately trigger immune responses. Not only do the authors show how the two helper complexes, together with either PAD4 or SAG101, contribute to resistance using genetic tools. The authors also use high quality interaction studies to prove the discrete complexes in vivo. This reviewer here certainly finds this manuscript interesting, well written and figures nicely presented. Immune receptors such as NLRs are common to most animal and plants. Thus, deep insight to how these complexes function in different organisms, with both similarities and differences, should be of interest to researchers in many fields. The particular question the authors address, the distinct function of EDS1 family members and plant helper NLRs, is of particular interest to the plant field.

While this happens very rarely, this reviewer thinks there is no reason to put any delay on a manuscript like this just to make sure authors are doing some more assays. Instead, this reviewer would like to thank the authors for giving me the pleasure to read some interesting Science in these chaotic times.

Reviewer #2 (Remarks to the Author):

Recently, there has been intense interest focusing on downstream signaling components of TNLs or CNLs. In particular, helper NLRs, like NRG1 and ADR1 have been intensely investigated and shown to be important for TNL signaling. In this manuscript, the authors provide detailed analyses of two lipase-like proteins known to be involved downstream of TNL signaling. They demonstrate that two higher order complexes exist, EDS1/SAG1/NRG1 that form upon effector recognition, and EDS1/PAD4/ADR1 whose formation does not require effector recognition. Previous published genetic data implicated these branches, this paper provides biochemical evidence and genetic evidence based on higher order mutants. Overall, the data in the paper seems solid, but I am missing raw data and magnitude of effects in the figures. I found the overall findings a bit minutia focusing on known genes that are already involved in TNL signaling. Also, the paper is not written in a style that is accessible to a broad audience in plant defense. I think only an expert in NLR signaling who has a strong background in the EDS1 family would be able to follow the paper past the introduction.

The figure displays do not help with interpretation either and are displayed in a nonstandard format, not showing raw data or trends but statistics only. Display of bacterial growth statistics in Fig 1 C, Figure 2a, and conductivity in 2b are difficult to interpret. Similar comments for MS protein abundance data in Fig 3.

The authors should be careful with imputation of protein values in figure 3 by MS. It really depends on how many missing values there are and the consistency if imputation would be correct. This also ignores variation that is present in the data.

Reviewer #3 (Remarks to the Author):

The manuscript by Sun et al. characterizes differential contributions by the EDS1-SAG101-NRG1s and EDS1-PAD4-ADR1s modules in basal immunity, TNL- and CNL-mediated ETI. The authors build on previous genetic evidence with a comprehensive analysis of different mutant combinations to show that there is no measurable cross-utilization of module components, i.e. SAG101 does not appear to utilize ADRs and PAD4 does not NRG1s. While some of this could be anticipated this careful dissection and documentation is extremely valuable for the field, Furthermore, the

manuscript demonstrates that this genetic finding rests on differential protein interactions between EDS1-SAG101 and NRG1s and EDS1-PAD4 and ADR1s, respectively. Interestingly, the authors discovered an effector recognition dependent step of complex formation between NRG1s and EDS1-SAG101 heterodimers, providing a novel molecular insight on ETI signaling following sensor resistance protein activation. Overall, this study is well designed and experiments are well presented. The authors in their introduction and discussion manage to pull several recent developments and their results into a succinct and yet comprehensive picture of ETI activation.

Below are only a few suggestions and concerns:

Page 8, lines 189-199: the authors discussed a possible role of SA in suppressing cell death. The authors tested mutant combinations of Metacaspase 1 (MC1) with SAG101-NRG1s and EDS1-PAD4-ADR1s modules. A recent study by the Dong lab demonstrated that SA promotes formation of NPR1 condensates to suppress CNL- and TNL-mediated cell death. This work should be included in this context.

Fig 3a and related text: the authors utilized pPAD4:YFP-PAD4 and pSAG101:SAG101-YFP in the pad4 sag101 double mutant background. The rationale for characterizing the PAD4 interactome in the absence of SAG101 and the SAG101 interactome in the absence of PAD4 is not clear and needs to be described. As the authors mentioned in line 430-431 "Hence, each TNL signaling branch has a transcriptional reprogramming and cell death-inducing capacity, dependent on the status of other pathways in the network", the proteomic data obtained in the double mutant background has potential limitations compared to complemented single mutant backgrounds. For example, the SAG101 interactome is being characterized in a genotype that displays reduced basal resistance. The EDS1 and NRG1 interactomes in contrast are determined in complemented single mutant lines with both the PAD4 and SAG101 sector functional. Thus, the authors should discuss the limitations and advantages for this design.

Minor comments:

1) On the face of it, it may be surprising for the less initiated reader that the manuscript mainly focuses on SAG101 when a sag101 mutant (or associated components) is demonstrated to not show reduced ETI whereas a pad4 mutant appears to (Fig. 1). SAG101 only seems to effect the HR response, a lack of which does not seem to impact bacterial growth. The authors may want to more explicitly emphasize results in Supplementary Fig. 2 showing that pad4, unlike sag101, is also more susceptible to virulent Pst to a similar extent as to Pst avrRps4, indicating that both PAD4 and SAG101 in parallel contribute to ETI in terms of inhibiting bacterial growth.

Line 18: superscript 7 not used in the author list.

Line 134 and rest of manuscript: Pf0-1 is a wildtype non-pathogenic *P. fluorescens* strain lacking a TTSS. This strain was used to engineer the strain called, perhaps not conveniently, EtHAn. It is better to be accurate though.

Lines 136-137: "same phenotypic clustering" yes, but a bit confusing without expanding that the phenotypic results are opposite (sag101 no HR but full basal resistance, pad4 the reverse).

Fig. 1c and other figures: it is recommended to assign letters to significance groups in descending order as is usual, not in discontinuous order. This is not so much a problem in bar graphs, but without relative context labels such as "ac" are confusing (the subtle color differences do not provide enough contrast to make clear that "b" is not included in "ac" etc.). Please clarify if assignments represent a comparison of everything to everything, not limited to columns.

Reviewer #1 (Remarks to the Author):

The manuscript by Sun and co-workers entitled "Pathogen effector recognition-dependent association of NRG1 with EDS1 and SAG101 in TNL receptor immunity" describes how two groups of helper NLRs in Arabidopsis form distinct complexes with specific EDS1 family members to appropriately trigger immune responses. Not only do the authors show how the two helper complexes, together with either PAD4 or SAG101, contribute to resistance using genetic tools. The authors also use high quality interaction studies to prove the discrete complexes in vivo.

This reviewer here certainly finds this manuscript interesting, well written and figures nicely presented. Immune receptors such as NLRs are common to most animal and plants. Thus, deep insight to how these complexes function in different organisms, with both similarities and differences, should be of interest to researchers in many fields. The particular question the authors address, the distinct function of EDS1 family members and plant helper NLRs, is of particular interest to the plant field.

While this happens very rarely, this reviewer thinks there is no reason to put any delay on a manuscript like this just to make sure authors are doing some more assays. Instead, this reviewer would like to thank the authors for giving me the pleasure to read some interesting Science in these chaotic times.

The authors thank the reviewer for the very positive appraisal of this work.

Reviewer #2 (Remarks to the Author):

Recently, there has been intense interest focusing on downstream signaling components of TNLs or CNLs. In particular, helper NLRs, like NRG1 and ADR1 have been intensely investigated and shown to be important for TNL signaling. In this manuscript, the authors provide detailed analyses of two lipase-like proteins known to be involved downstream of TNL signaling. They demonstrate that two higher order complexes exist, EDS1/SAG1/NRG1 that form upon effector recognition, and EDS1/PAD4/ADR1 whose formation does not require effector recognition. Previous published genetic data implicated these branches, this paper provides biochemical evidence and genetic evidence based on higher order mutants. Overall, the data in the paper seems solid, but I am missing raw data and magnitude of effects in the figures. I found the overall findings a bit minutia focusing on known genes that are already involved in TNL signaling. Also, the paper is not written in a style that is accessible to a broad audience in plant defense. I think only an expert in NLR signaling who has a strong background in the EDS1 family would be able to follow the paper past the introduction.

We thank the reviewer for the helpful comments. We now provide a full inventory of the original data sets used for Figs 1-3, showing spread of the data points (please see specific points below). Where possible, we explain and clarify terms in the text for a more general plant defence reader while keeping the central strand and focus of the analysis.

The figure displays do not help with interpretation either and are displayed in a nonstandard format, not showing raw data or trends but statistics only. Display of bacterial growth statistics in Fig 1 C, Figure 2a, and conductivity in 2b are difficult to interpret. Similar comments for MS protein abundance data in Fig 3.

Thank you for pointing this out. We reflected on the clearest way to represent extensive bacterial growth and electrolyte leakage data of the different mutants in a visually attractive and easy-to-digest form. The data in main Figs 1 and 2 are integral to the study since they provide solid genetic evidence that PAD4 and SAG101 do not utilize NRG1 and ADR1 group RNLs, respectively, for effective defence. We are confident that showing all pathogen growth and electrolyte leakage assays in Figs 1 and 2 in the format of jitter, barplots or boxplots, as in Fig. 1c, would not be effective data presentation. We take on board fully the reviewer's

request to show the corresponding original data variation beyond the statistics. Therefore, in addition to displaying the bacterial growth data in Supplemental Fig. 2a – f, we provide boxplots in revised Supplemental Figs 1f, 1g, 2g, 2h for all electrolyte leakage assays shown in the manuscript.

Similarly, we display mass spectrometry (MS) data as relative abundance estimates for proteins (label-free quantification (LFQ) values) as jitter plots rather than heatmaps in revised main Fig. 3a and 3b. We thank the reviewer for the suggestion to try a different presentation mode. The dot plot representation in revised Fig. 3a and 3b is more intuitively intelligible, and it demonstrates the copurification of RNL groups with PAD4, SAG101 and EDS1 more accurately and convincingly. At the same time, we believe the heatmap representation of MS data in Supplemental Figure 5b is more compact and therefore remains unchanged. Since source and processed MS data are provided in the PRIDE database and Supplemental Tables 4, 5 and 6 show LFQ intensities and peptide count values per replicate, interested readers have sufficient information to form their own impression about the spread of individual datapoints for any protein detected in our MS experiments.

The authors should be careful with imputation of protein values in figure 3 by MS. It really depends on how many missing values there are and the consistency if imputation would be correct. This also ignores variation that is present in the data.

Missing values are common in all 'omics datasets, for instance, due to a lack of the molecular feature in a sample or due to the instrument detection limit. The statistical analysis of proteomic label-free quantitation data is virtually impossible without imputation, since ratio calculations against 0 are impossible. For MS data analysis, protein groups were filtered to have at least three values in at least one group and the imputed missing values were based on the minimum detected protein abundances in the sample, to retain a normally distributed dataset for statistical analyses (Lazar et al., 2016). We added the following sentence in the Supplemental Materials and Methods to clarify: **“The imputed missing values were based on the minimum detected protein abundances in the sample to retain a normally distributed dataset for statistical analyses (Tyanova et al., 2016) (1.8 downshift, separately for each column).”**. With these assumptions, we are confident that missing values were imputed in an appropriate manner. In the revised Supplemental Tables 4 and 5, we provide LFQ values before and after imputation. The Tables with imputed LFQ values were used to perform statistical analysis and to generate volcano plots (Supplemental Figures 5a and 6), which are essential for appropriate MS data representation.

Reviewer #3 (Remarks to the Author):

The manuscript by Sun et al. characterizes differential contributions by the EDS1-SAG101-NRG1s and EDS1-PAD4-ADR1s modules in basal immunity, TNL- and CNL-mediated ETI. The authors build on previous genetic evidence with a comprehensive analysis of different mutant combinations to show that there is no measurable cross-utilization of module components, i.e. SAG101 does not appear to utilize ADRs and PAD4 does not NRG1s. While some of this could be anticipated this careful dissection and documentation is extremely valuable for the field, Furthermore, the manuscript demonstrates that this genetic finding rests on differential protein interactions between EDS1-SAG101 and NRG1s and EDS1-PAD4 and ADR1s, respectively. Interestingly, the authors discovered an effector recognition dependent step of complex formation between NRG1s and EDS1-SAG101 heterodimers, providing a novel molecular insight on ETI signaling following sensor resistance protein activation. Overall, this study is well designed and experiments are well presented. The authors in their introduction and discussion manage to pull several recent developments and their results into a succinct and yet comprehensive picture of ETI activation.

We thank the reviewer for these supportive comments and specifically for highlighting the importance of genetic and molecular evidence shown in the manuscript.

Below are only a few suggestions and concerns:

Page 8, lines 189-199: the authors discussed a possible role of SA in suppressing cell death. The authors tested mutant combinations of Metacaspase 1 (MC1) with SAG101-NRG1s and EDS1-PAD4-ADR1s modules. A recent study by the Dong lab demonstrated that SA promotes formation of NPR1 condensates to suppress CNL- and TNL-mediated cell death. This work should be included in this context.

*Thank you for reminding us of this relevant reference. The following sentence was added to the respective Discussion section: “**This aligns with a reported role of SA receptor non-expressor of PR1 (NPR1) in limiting NLR cell death in the presence of high SA concentrations around the tissue undergoing cell death (Zavaliev et al., 2020)**”.*

Fig 3a and related text: the authors utilized pPAD4:YFP-PAD4 and pSAG101:SAG101-YFP in the pad4 sag101 double mutant background. The rationale for characterizing the PAD4 interactome in the absence of SAG101 and the SAG101 interactome in the absence of PAD4 is not clear and needs to be described. As the authors mentioned in line 430-431 “Hence, each TNL signaling branch has a transcriptional reprogramming and cell death-inducing capacity, dependent on the status of other pathways in the network”, the proteomic data obtained in the double mutant background has potential limitations compared to complemented single mutant backgrounds. For example, the SAG101 interactome is being characterized in a genotype that displays reduced basal resistance. The EDS1 and NRG1 interactomes in contrast are determined in complemented single mutant lines with both the PAD4 and SAG101 sector functional. Thus, the authors should discuss the limitations and advantages for this design.

*Thanks to the reviewer for pointing this out. We add a sentence in this section of Results to clarify: “**The double pad4 sag101 mutant background of the pSAG101:SAG101-YFP complementation line allows SAG101-NRG1s to engage in ETI cell death and pathogen resistance conferred by RRS1-RPS4 ((Lapin et al., 2019), Supplementary Fig. 4) and in transcriptional reprogramming that is otherwise chiefly controlled by PAD4-ADR1s in RRS1-RPS4 ETI (Bhandari et al., 2019; Saile et al., 2020)**”. We further add “**The dynamics of EDS1-SAG101-NRG1s associations relative to presumed EDS1-PAD4-ADR1s associations in immune-activated cells and tissues remain unresolved**” in the section of Discussion devoted to IP-MS data.*

Minor comments:

1) On the face of it, it may be surprising for the less initiated reader that the manuscript mainly focuses on SAG101 when a sag101 mutant (or associated components) is demonstrated to not show reduced ETI whereas a pad4 mutant appears to (Fig. 1). SAG101 only seems to effect the HR response, a lack of which does not seem to impact bacterial growth. The authors may want to more explicitly emphasize results in Supplementary Fig. 2 showing that pad4, unlike sag101, is also more susceptible to virulent Pst to a similar extent as to Pst avrRps4, indicating that both PAD4 and SAG101 in parallel contribute to ETI in terms of inhibiting bacterial growth.

*The reviewer is correct that the SAG101/NRG1 branch does not contribute much to Pseudomonas resistance when PAD4/ADR1 are present. (Castel et al., 2019) showed that SAG101 and NRG1 make a greater contribution to resistance to the obligate biotrophic pathogen Hpa. Hence the SAG101/NRG1 module (and its cell death promoting activity) appears to be more important in resistance to haustorial biotrophic pathogens. To strengthen the point that SAG101 is engaged in both resistance and cell death in TNL ETI, we added a sentence at the end of Results section describing the discovery of EDS1 family – RNL associations (Fig. 3): “**The observed preferential association of PAD4 with ADR1-L1 and***

ADR1-L2, and SAG101 with NRG1.1 and NRG1.2, further suggests that EDS1-PAD4 and EDS1-SAG101 associations with specific helper RNL types underpin these genetically distinct Arabidopsis immunity modules that can promote resistance and cell death responses in TNL ETI ((Castel et al., 2019; Lapin et al., 2019), Fig. 1, Fig. 2)".

Line 18: superscript 7 not used in the author list.

We corrected the affiliation of Iris Finkemeier. Thanks for pointing out.

Line 134 and rest of manuscript: Pf0-1 is a wildtype non-pathogenic *P. fluorescens* strain lacking a TTSS. This strain was used to engineer the strain called, perhaps not conveniently, EtHAn. It is better to be accurate though.

*Thanks for pointing this out. We would like to keep to Pf0-1 because it is used in key cited papers here and EtHAn as the name for this strain has been very unevenly adopted in the field. We write in the text at first mention in Results ‘type III secretion system-equipped *P. fluorescens* strain Pf0-1’. We now have in Methods ‘For cell death assays, a type III secretion system-equipped *Pseudomonas fluorescens* effector tester strain Pf0-1 avrRps4 (Thomas et al., 2009) was resuspended in 10 mM MgCl₂ (OD₆₀₀=0.2) and syringe-infiltrated into leaves. Only the type III secretion system-equipped and not the wild type Pf0-1 strain was used in this study, and for simplicity we refer to it as “Pf0-1”’.*

Lines 136-137: "same phenotypic clustering" yes, but a bit confusing without expanding that the phenotypic results are opposite (sag101 no HR but full basal resistance, pad4 the reverse).

Thanks, we expanded the sentence to avoid ambiguity as suggested. The new sentence in Results is: “This produced the same phenotypic clustering of mutants as the Pst avrRps4 resistance assays – pad4 and a3 behaved like pad4 a3, while defects in sag101 and n2 aligned with those in sag101 n2”.

Fig. 1c and other figures: it is recommended to assign letters to significance groups in descending order as is usual, not in discontinuous order. This is not so much a problem in bar graphs, but without relative context labels such as "ac" are confusing (the subtle color differences do not provide enough contrast to make clear that "b" is not included in "ac" etc.). Please clarify if assignments represent a comparison of everything to everything, not limited to columns.

Authors thank the reviewer for this advice. The significance codes are now reassigned on all plots, so “a” refers to the lowest value in the range of mean phenotypic values on a given plot. On heatmaps in Fig. 1c, 2a, 2b, Supplemental Fig 1d and 1e, we add lines next to significance letter codes to guide interpretation of plots and reduce ambiguity. Barplots and boxplots representing the full extent of the data are now included as extended Supplemental Fig.1 and 2. In figure captions for Fig. 1, 2 and Supplemental Fig. 1, we added the following statement to help reading of the plots: “Note: significance codes are assigned based on the statistical analysis per treatment or timepoint and should be read columnwise”.

Bhandari, D.D., Lapin, D., Kracher, B., von Born, P., Bautor, J., Niefind, K., and Parker, J.E. (2019). An EDS1 heterodimer signalling surface enforces timely reprogramming of immunity genes in Arabidopsis. *Nature communications* **10**, 772.

Castel, B., Ngou, P.M., Cevik, V., Redkar, A., Kim, D.S., Yang, Y., Ding, P., and Jones, J.D.G. (2019). Diverse NLR immune receptors activate defence via the RPW8-NLR NRG1. *The New phytologist* **222**, 966-980.

- Lapin, D., Kovacova, V., Sun, X., Dongus, J.A., Bhandari, D., von Born, P., Bautor, J., Guarneri, N., Rzemieniewski, J., Stuttmann, J., Beyer, A., and Parker, J.E.** (2019). A Coevolved EDS1-SAG101-NRG1 Module Mediates Cell Death Signaling by TIR-Domain Immune Receptors. *The Plant cell* **31**, 2430-2455.
- Lazar, C., Gatto, L., Ferro, M., Bruley, C., and Burger, T.** (2016). Accounting for the Multiple Natures of Missing Values in Label-Free Quantitative Proteomics Data Sets to Compare Imputation Strategies. *Journal of proteome research* **15**, 1116-1125.
- Saile, S.C., Jacob, P., Castel, B., Jubic, L.M., Salas-González, I., Bäcker, M., Jones, J.D.G., Dangl, J.L., and El Kasm, F.** (2020). Two unequally redundant "helper" immune receptor families mediate *Arabidopsis thaliana* intracellular "sensor" immune receptor functions. *PLoS biology* **18**, e3000783.
- Thomas, W.J., Thireault, C.A., Kimbrel, J.A., and Chang, J.H.** (2009). Recombineering and stable integration of the *Pseudomonas syringae* pv. *syringae* 61 hrp/hrc cluster into the genome of the soil bacterium *Pseudomonas fluorescens* Pf0-1. *The Plant journal : for cell and molecular biology* **60**, 919-928.
- Tyanova, S., Temu, T., Sinitcyn, P., Carlson, A., Hein, M.Y., Geiger, T., Mann, M., and Cox, J.** (2016). The Perseus computational platform for comprehensive analysis of (prote)omics data. *Nature Methods* **13**, 731-740.
- Zavaliev, R., Mohan, R., Chen, T., and Dong, X.** (2020). Formation of NPR1 Condensates Promotes Cell Survival during the Plant Immune Response. *Cell* **182**, 1093-1108.e1018.

REVIEWERS' COMMENTS

Reviewer #1 (Remarks to the Author):

This reviewer was satisfied with the first version of the ms and this second is substantially improved. Thus, this reviewer has no further comments.

Reviewer #3 (Remarks to the Author):

The first version was already in great shape, and the authors addressed all my comments. Regarding the information added after resubmission, the added information in lines 469-476 is acceptable and informative, and in my mind the authors are correct in stating that the topic of clarification is beyond the scope of/not overlapping with this manuscript.

I appreciate the authors' reflections on how to present a large amount of data in a digestible format. Perhaps it's just me but reassigning significance letters makes the heatmaps more intuitively understandable. Just two minor suggestions to complete the reassignments:

-Fig 1 (c) avrRpt2 column and Fig. S1 (a): assign "a" to the n2, sag101, sag101 n2 block, leave Col at "ab", change the two "ac" further down to "bc". You could add a row for eds1 to the heatmap.

-Similarly in Fig 2 (a) column EV: assign "a" to n2 sag101 block, change "ac" in 5th block to "bc", consistent with Fig S2 (e) and (f).

Walter Gassmann

Response to referees

Reviewer #1 (Remarks to the Author):

This reviewer was satisfied with the first version of the ms and this second is substantially improved. Thus, this reviewer has no further comments.

Response

We thank the reviewer for checking our revised manuscript.

Reviewer #3 (Remarks to the Author):

The first version was already in great shape, and the authors addressed all my comments. Regarding the information added after resubmission, the added information in lines 469-476 is acceptable and informative, and in my mind the authors are correct in stating that the topic of clarification is beyond the scope of/not overlapping with this manuscript.

I appreciate the authors' reflections on how to present a large amount of data in a digestible format. Perhaps it's just me but reassigning significance letters makes the heatmaps more intuitively understandable. Just two minor suggestions to complete the reassignments:

-Fig 1 (c) avrRpt2 column and Fig. S1 (a): assign "a" to the n2, sag101, sag101 n2 block, leave Col at "ab", change the two "ac" further down to "bc". You could add a row for eds1 to the heatmap.

-Similarly in Fig 2 (a) column EV: assign "a" to n2 sag101 block, change "ac" in 5th block to "bc", consistent with Fig S2 (e) and (f).

Walter Gassmann

Response

Dear Walter, thank you for carefully looking at our revised manuscript. We followed your advice to simplify and further improve assignments of the significance codes. We also added the row for eds1 in Fig 1c as suggested. Thanks for pointing out this inconsistency. Kind regards, Authors.